# Thinking with Camera: A Unified Multimodal Model for Camera-Centric Understanding and Generation

**Kang Liao**[1]   **Size Wu**[1]   **Zhonghua Wu**[2]   **Linyi Jin**[3]
**Chao Wang**[4]   **Yikai Wang**[1]   **Fei Wang**[2]   **Wei Li**[1 †]   **Chen Change Loy**[1 †]

[1]S-Lab, Nanyang Technological University   [2]SenseTime Research
[3]University of Michigan   [4]Max-Planck Institute for Informatics

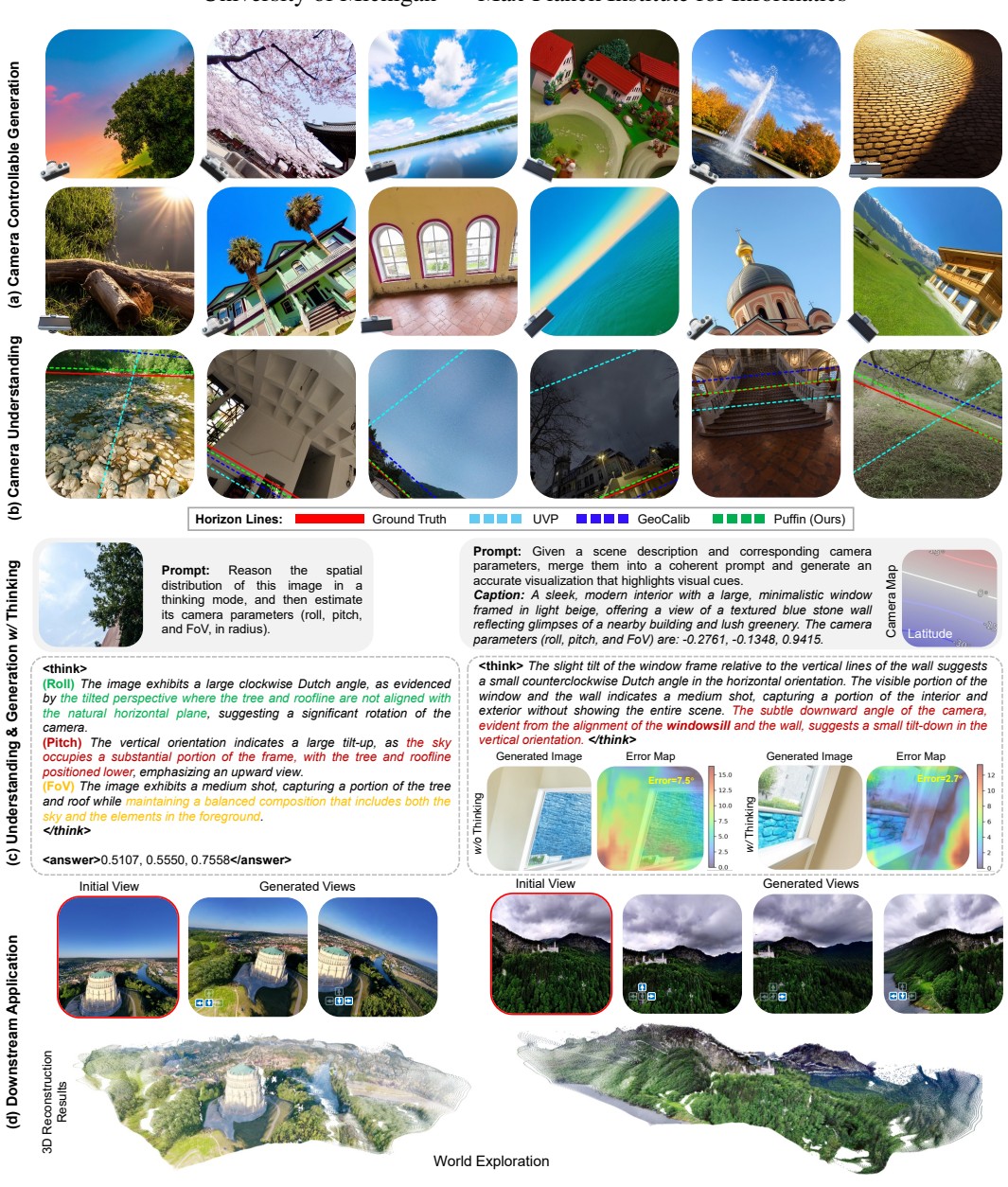

**Figure 1: Illustration of the versatile capabilities of our Puffin model. It unifies camera-centric generation (a) and understanding (b), supports the thinking mode (c), and enables diverse 3D applications (d).**

---

[†] Corresponding authors

## ABSTRACT

Camera-centric understanding and generation are two cornerstones of spatial intelligence, yet they are typically studied in isolation. We present **Puffin**, a unified camera-centric multimodal model that extends spatial awareness along the camera dimension. Puffin integrates language regression and diffusion-based generation to interpret and create scenes from arbitrary viewpoints. To bridge the modality gap between cameras and vision-language, we introduce a novel paradigm that treats *camera as language*, enabling *thinking with camera*. This guides the model to align spatially grounded visual cues with photographic terminology while reasoning across geometric context. Puffin is trained on **Puffin-4M**, a large-scale dataset of 4 million vision-language-camera triplets. We incorporate both global camera parameters and pixel-wise camera maps, yielding flexible and reliable spatial generation. Experiments demonstrate Puffin's superior performance over specialized models for camera-centric generation and understanding. With our designed instruction tuning, Puffin generalizes to diverse cross-view tasks such as spatial imagination, world exploration, and photography guidance. To advance multimodal spatial intelligence research, we release the code, models, dataset pipeline, and benchmark at https://kangliao929.github.io/projects/puffin.

## 1 INTRODUCTION

For machines, cameras serve as the primary interface to the physical world, enabling spatial intelligence that underlies applications such as robotics, AR/VR, and autonomous driving. In general, two principal camera-centric objectives work in tandem to enable machines to perceive and interact with their spatial context. On the one hand, *understanding* the camera geometry from images (Pollefeys et al., 1999; Veicht et al., 2024; Zhang et al., 2024; Lin et al., 2025c), namely how the 3D world is projected onto the 2D image plane, lays the foundation for machines to recover spatial structure and navigate complex environments. On the other hand, by modulating intrinsic and extrinsic parameters, cameras encode spatial relationships and offer flexible control over spatial content *generation* (Bernal-Berdun et al., 2025; He et al., 2024; Ren et al., 2025; Ball et al., 2025), which simulates how the world appears from any viewpoint or orientation. To date, these two perspectives have been commonly treated as isolated problems and independently explored by the research community.

In this work, we make *the first attempt* to unify camera-centric understanding and generation in a cohesive framework. Motivated by recent progress in unified understanding and generation with large multimodal models (LMMs) (Team, 2024; Wu et al., 2025d; Pan et al., 2025; Wu et al., 2025b), we extend this paradigm to the spatial domain, where camera geometry plays a central role. However, unlike language or images, camera parameters are abstract and non-intuitive: they describe field-of-view (FoV), orientation, or perspective in numerical form rather than semantic content. This discrepancy introduces a modality gap when integrating cameras into LMMs. For instance, when users specify "20° roll" or "35mm lens" for controllable generation, existing models often ignore or misinterpret such cues, pursuing semantic alignment while neglecting precise spatial control. Similarly, current LMMs tend to collapse geometric details into coarse representations when understanding camera information, leading to spatially inconsistent outputs. As a result, naïvely extending LMMs cannot resolve conflicts between modalities, producing suboptimal performance in both tasks.

To address this challenge, we introduce **Puffin**, a unified multimodal framework that interprets cameras as a first-class modality. Puffin combines autoregressive and diffusion modeling to jointly perform camera-centric understanding and generation[1]. Instead of treating camera parameters as auxiliary labels, Puffin introduces the notion of *thinking with camera*, aligning spatially grounded visual cues with professional photographic terminology while reasoning over geometric context. This design provides a shared chain-of-thought across multimodal tasks, enabling spatially consistent understanding and controllably aligned generation.

---

[1]We mainly focus on single-view calibration and text-to-image controllable generation, but Puffin can be flexibly extended to cross-view understanding and generation via our designed instruction tuning (see Figure 6).

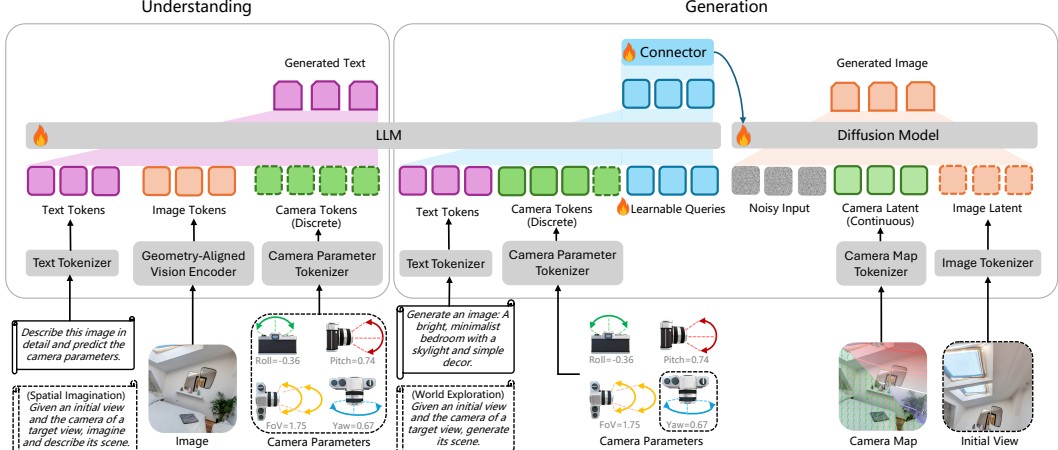

**Figure 2: Overview of the proposed Puffin.** It jointly learns the camera-centric understanding and generation tasks in a unified multimodal framework. The elements bounded with dotted boundaries represent the cross-view understanding and generation during instruction tuning, such as spatial imagination and world exploration.

To support this framework, we construct **Puffin-4M**, a large-scale dataset of 4 million vision-language-camera triplets. Puffin-4M includes single-view images with precise camera parameters, descriptive captions, pixel-wise camera maps, and spatial reasoning annotations across diverse indoor and outdoor scenarios. Beyond single views, it also incorporates cross-view and aesthetic images, making it a versatile benchmark for both understanding and generation tasks.

Experimental results show Puffin outperforms specialized models for camera-centric understanding or generation, and can be adapted to diverse downstream applications. We illustrate the versatile capabilities of our Puffin model in Figure 1. In each generated image (a), the target camera is marked at the bottom left, and the horizon lines are visualized from the estimated camera parameters (b). For world exploration (d), we visualize 3D reconstruction results derived from the initial and generated views. Our main contributions are threefold:

- We make *the first attempt* to seamlessly integrate camera geometry into a unified multimodal model, introducing a camera-centric framework to advance multimodal spatial intelligence.

- We propose *thinking with camera*, a novel mechanism that guides the model to align spatially grounded visual cues with photographic terminology, bridging the modality gap between camera and vision-language and enabling effective spatial reasoning.

- We construct **Puffin-4M**, a large-scale dataset of 4M vision-language-camera triplets spanning diverse indoor and outdoor scenes, and establish a comprehensive benchmark for evaluating camera-centric multimodal models.

## 2 CAMERA-CENTRIC UNIFIED MULTIMODAL MODEL

Puffin, as illustrated in Figure 2, unifies camera-centric understanding and generation within a multimodal paradigm. For understanding, we introduce a geometry-aligned vision encoder to a large language model (LLM) to retain rich geometric features and enhance the model's capacity for spatial analysis. For generation, a connector module learns to map the hidden states of the LLM (via a set of learnable queries) into conditioning signals that can be interpreted by the diffusion model. To facilitate the integration of camera geometry, apart from the discrete camera tokens derived from numerical camera parameters, we introduce continuous camera latent obtained from pixel-wise camera maps, allowing fine-grained spatial control in image generation.

### 2.1 CAMERA UNDERSTANDING

**Definition.** In this work, camera understanding is formulated as a question-answering task conditioned on image content. The generated text consists of a concise description or spatial reasoning along with the estimated camera parameters (*i.e.*, roll, pitch, FoV) of the input image. Unlike previous methods

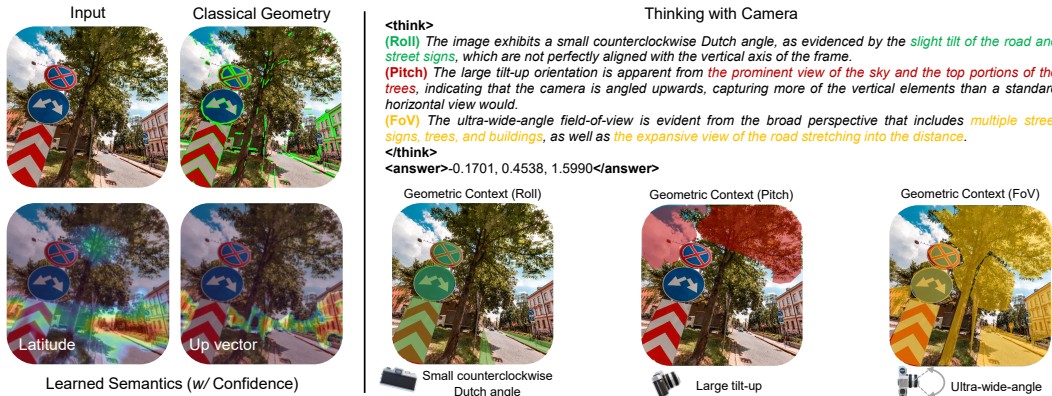

**Figure 3: Methods for learning camera geometry.** (Left) Previous classical and learning-based methods focused on extracting or learning representations such as geometric structures or semantic features (with confidence). (Right) We introduce the notion of *thinking with camera* through LMMs. It first decouples the camera parameters across geometric context, establishing connections between spatially grounded visual cues (highlighted in the masked regions) and professional photographic terms. The camera parameters are then predicted within the **<answer></answer>** tag through this spatial reasoning process **<think></think>**.

that directly estimate the parameters from images, our approach integrates camera geometry within the text and performs next-token prediction in a multimodal sequence modeling paradigm.

**Motivation.** As illustrated in Figure 3 (left), previous classical and learning-based methods focus on extracting or learning representations to predict the camera parameters, such as geometric structures (Pautrat et al., 2023) or semantic features with confidence estimates (Veicht et al., 2024). However, these representations often emphasize low-/mid-level patterns, limiting their ability to capture a holistic and coherent spatial concept. As a result, existing approaches tend to excel in scenarios with rich features but struggle to generalize across diverse visual environments.

**Thinking.** Instead of focusing on how to learn a representation, we propose to interpret the camera as language and introduce the notion of *thinking with camera*. It guides the LMMs to align spatially grounded visual cues with photographic terminology while reasoning across geometric context. The details of each key element are elaborated below.

• *Spatially Grounded Visual Cues.* The 3D world is governed by physical laws, where gravity and human design shape stable spatial regularities that serve as strong perceptual priors. Texture-less regions such as sky, ceilings, floors, or ground surfaces lack local features but encode vertical regularities critical for pitch estimation. Similarly, FoV estimation relies on perceiving spatial composition, including the foreground–background ratio, object scale, and depth distribution. While such properties are difficult to infer from purely visual representations, they are implicitly captured by LMMs as knowledge priors. Thus, we embed these spatially grounded visual cues into our thinking captions, enabling the model to perform explicit spatial reasoning about camera geometry.

• *Professional Photographic Terms.* Existing LMMs typically acquire over-abstracted semantics, whereas the detailed numerical values of camera parameters are too fine-grained to estimate precisely. As a practical alternative, professional photographic terms (*e.g.*, close-up, tilt-up, Dutch angle) are widely used in annotations and well aligned with LMM knowledge (Liu et al., 2025; Wang et al., 2025b; Lin et al., 2025c). Thus, we leverage them as intermediate supervisory signals to naturally bridge low-/mid-level camera geometry and high-level multimodal reasoning. These terms, derived as quantized abstractions of camera parameters, are merged with textual scene descriptions, making global spatial arrangements linguistically accessible. The parameter-to-term mapping can be formulated as $f : p \mapsto t$, in which the mapping $f$ is shown in Appendix A2.2 (Table A1).

• *Geometric Context.* As shown in Figure 3 (right), we decouple camera parameters across geometric context (roll, pitch, and FoV), which aligns specific spatially grounded visual cues such as sky, foreground composition, and object-level depth ordering with each professional photographic terminology. By anchoring numerical attributes to semantically meaningful descriptors, our framework bridges abstract visual features and physically interpretable geometry. The final parameters are predicted through this structured spatial reasoning.

With the above designs, we interpret the camera as language by grounding its physical attributes in stable spatial regularities. Numerical parameters are abstracted into professional photographic terms, providing a semantic vocabulary aligned with LMMs. Through this mapping, camera geometry becomes linguistically interpretable, allowing structured spatial reasoning for accurate camera parameter prediction. We visualize more reasoning results in Appendix A5.1 (Figure A8).

**Choosing a Suitable Vision Encoder.** A straightforward approach to camera understanding is to fine-tune existing LMMs that couple a vision encoder with an LLM, but this naïve strategy faces two major limitations: (i) vision encoders in LMMs are primarily designed for recognition tasks and thus yield condensed features lacking geometric fidelity, and (ii) language components contain little prior knowledge of spatial perception, reducing adaptability to camera-centric tasks. As a result, such fine-tuning can lead to performance bottlenecks and even underperform pure vision-based methods (see Section 3.3). To overcome these issues, we introduce a *geometry-aligned vision encoder* distilled from both semantic (*e.g.*, CLIP, SigLIP) and vision-centric (*e.g.*, DINO, SAM) teachers (Heinrich et al., 2025), offering versatile features that preserve geometric fidelity while maintaining strong semantic understanding. We then align this encoder with an LLM (Qwen et al., 2024) via progressive unfreezing and joint fine-tuning. This staged optimization stabilizes training and fosters spatial awareness that bridges low-/mid-level structural cues with high-level linguistic reasoning. The detailed training recipe is provided in Appendix A4.

## 2.2 CAMERA-CONTROLLABLE GENERATION

**Motivation.** Unlike image understanding, image generation requires complex cross-modal alignment and the synthesis of fine-grained visual details. As discussed in Section 2.1, the detailed numerical values of camera parameters are too specific for current LMMs to interpret effectively, failing to faithfully capture the realistic spatial distribution required for camera-controllable generation.

**Thinking.** To address this, we design a step-by-step process that integrates visual detail analysis with reasoning. The model first infers the potential visual cues from vanilla captions, and then uses this textual reasoning as a semantic planning stage to guide image generation. For instance, a large pitch value may correspond to an expansive sky with clouds in outdoor scenes or to pendant lights and uncluttered ceilings indoors. Beyond textual reasoning, numerical camera parameters are translated into professional photographic terms more suitable for LMMs, naturally aligning with the reasoning process in camera understanding. We therefore adopt a shared chain-of-thought mechanism between understanding and controllable generation. As shown in Figure 1 (c), given a small pitch value and a caption describing a modern interior, our method translates the value into a photographic term (*e.g.*, small tilt-down), imagines salient cues such as a windowsill, and produces more precise spatial simulation than the baseline.

**Flexible and Faithful Control.** The pipeline of camera-controllable generation is shown in Figure 2 (right). The key design is to incorporate pixel-wise camera maps as a continuous latent of camera geometry, apart from the discrete camera tokens derived from numerical parameters. Unlike tokens that capture only global attributes, these dense maps encode local geometric context at each pixel, including orientation and displacement cues (Jin et al., 2023). By converting maps into continuous latent, the diffusion model receives fine-grained spatial priors that preserve global camera settings while adapting to subtle geometric variations, thus offering flexible control of spatial layout and viewpoint. Additionally, we introduce a connector module as an adaptive interface between the LLM and the diffusion model, where a set of learnable queries together with text and camera tokens extract and restructure LLM hidden representations, which are then projected into conditioning signals for generation Pan et al. (2025); Wu et al. (2025c). This design enables semantic and geometric understanding from the LLM to faithfully guide the diffusion model.

## 2.3 INSTRUCTION TUNING

Although our Puffin focuses on single-view camera calibration and text-to-image controllable generation, it can be flexibly extended to cross-view settings with only minor modifications, such as appending additional tokens and switching prompts according to the target task. As shown in Figure 2, the dotted modules denote cross-view understanding and generation. We explore three tasks: (i) spatial imagination, where the model imagines the scene description of a target view given its camera parameters and an initial view; (ii) world exploration, where the model generates the target view,

**Table 1: Evaluation results on camera understanding.** We color the best and second best results.

| | Approach | Roll [degrees] | | | | Pitch [degrees] | | | | FoV [degrees] | | | |
|---|---|---|---|---|---|---|---|---|---|---|---|---|---|
| | | error ↓ | AUC ▷ 1/5/10° ↑ | | | error ↓ | AUC ▷ 1/5/10° ↑ | | | error ↓ | AUC ▷ 1/5/10° ↑ | | |
| **MegaDepth** | DeepCalib (Lopez et al., 2019) | 1.41 | 34.6 | 65.4 | 79.4 | 5.19 | 11.9 | 27.8 | 44.8 | 11.14 | 5.6 | 12.1 | 22.9 |
| | CTRL-C (Lee et al., 2021) | 0.88 | 54.5 | 75.0 | 84.2 | 4.80 | 16.6 | 33.2 | 46.5 | 18.65 | 2.0 | 5.8 | 12.8 |
| | MSCC (Song et al., 2024) | 0.90 | 53.1 | 72.8 | 82.1 | 5.73 | 19.0 | 33.2 | 44.3 | 10.80 | 6.0 | 14.6 | 26.2 |
| | ParamNet (Jin et al., 2023) | 1.17 | 43.4 | 70.7 | 82.2 | 3.99 | 15.4 | 34.5 | 53.3 | 11.01 | 3.2 | 10.1 | 21.3 |
| | SVA (Lochman et al., 2021) | - | 31.9 | 35.0 | 36.2 | - | 13.6 | 20.6 | 24.9 | - | 9.4 | 16.1 | 21.1 |
| | UVP (Pautrat et al., 2023) | 0.51 | 69.2 | 81.6 | 86.9 | 4.59 | 21.6 | 36.2 | 47.4 | 10.92 | 8.2 | 18.7 | 29.8 |
| | GeoCalib (Veicht et al., 2024) | 0.36 | 82.6 | 90.6 | 94.0 | 1.94 | 32.4 | 53.3 | 67.5 | 4.46 | 13.6 | 31.7 | 48.2 |
| | **Puffin (Ours)** | **0.32** | **84.9** | **93.4** | **96.2** | **1.08** | **47.6** | **68.2** | **79.4** | **2.42** | **23.9** | **47.8** | **64.1** |
| **TartanAir** | DeepCalib (Lopez et al., 2019) | 1.95 | 24.7 | 55.4 | 71.5 | 3.27 | 16.3 | 38.8 | 58.5 | 8.07 | 1.5 | 8.8 | 27.2 |
| | CTRL-C (Lee et al., 2021) | 1.68 | 32.8 | 59.1 | 74.1 | 2.39 | 24.6 | 48.6 | 65.2 | 5.64 | 10.7 | 25.4 | 43.5 |
| | MSCC (Song et al., 2024) | 3.50 | 15.0 | 37.2 | 57.7 | 3.48 | 18.8 | 38.6 | 54.3 | 11.18 | 4.4 | 11.8 | 23.0 |
| | ParamNet (Jin et al., 2023) | 1.63 | 34.5 | 59.2 | 73.9 | 3.05 | 19.4 | 42.0 | 60.3 | 8.21 | 6.0 | 16.8 | 31.6 |
| | SVA (Lochman et al., 2021) | 9.48 | 32.4 | 39.6 | 44.1 | 18.46 | 21.2 | 28.8 | 34.5 | 43.01 | 8.8 | 16.1 | 21.6 |
| | UVP (Pautrat et al., 2023) | 0.89 | 52.1 | 64.8 | 71.9 | 2.48 | 36.2 | 48.8 | 58.6 | 9.15 | 15.8 | 25.8 | 35.7 |
| | GeoCalib (Veicht et al., 2024) | 0.43 | 71.3 | 83.8 | 89.8 | 1.49 | 38.2 | 62.9 | 76.6 | **4.90** | 14.1 | **30.4** | **47.6** |
| | **Puffin (Ours)** | **0.40** | **71.7** | **86.2** | **92.1** | **0.95** | **51.0** | **68.2** | **79.3** | 7.48 | **16.3** | 28.5 | 39.0 |
| **LaMAR** | DeepCalib (Lopez et al., 2019) | 1.15 | 44.1 | 73.9 | 84.8 | 4.68 | 10.8 | 28.3 | 49.8 | 10.93 | 0.7 | 13.0 | 24.0 |
| | CTRL-C (Lee et al., 2021) | 1.20 | 43.5 | 70.9 | 82.5 | 1.94 | 27.0 | 54.7 | 70.2 | 5.64 | 9.8 | 24.6 | 43.2 |
| | MSCC (Song et al., 2024) | 1.44 | 39.6 | 60.7 | 72.8 | 3.02 | 20.9 | 41.8 | 55.7 | 14.78 | 3.2 | 8.3 | 16.8 |
| | ParamNet (Jin et al., 2023) | 0.93 | 51.7 | 77.0 | 86.0 | 2.15 | 27.0 | 52.7 | 70.2 | 14.71 | 2.8 | 6.8 | 14.3 |
| | SVA (Lochman et al., 2021) | - | 8.6 | 9.2 | 9.7 | - | 3.4 | 5.7 | 7.0 | - | 1.2 | 2.7 | 4.1 |
| | UVP (Pautrat et al., 2023) | 0.38 | 72.7 | 81.8 | 85.7 | 1.34 | 42.3 | 59.9 | 69.4 | 5.57 | 15.6 | 30.6 | 43.5 |
| | GeoCalib (Veicht et al., 2024) | **0.28** | **86.4** | **92.5** | **95.0** | 0.87 | 55.0 | 76.9 | 86.2 | **3.03** | **19.1** | **41.5** | **60.0** |
| | **Puffin (Ours)** | 0.38 | 80.6 | 89.8 | 93.5 | **0.71** | **61.7** | **78.9** | **86.4** | 3.62 | 17.0 | 37.3 | 53.1 |
| **Puffin-Und** | DeepCalib (Lopez et al., 2019) | 1.90 | 29.3 | 56.2 | 71.7 | 3.71 | 15.3 | 36.0 | 54.9 | 7.43 | 9.0 | 19.4 | 34.8 |
| | CTRL-C (Lee et al., 2021) | 4.69 | 20.3 | 35.2 | 46.7 | 8.43 | 10.8 | 24.6 | 36.1 | 11.70 | 5.3 | 12.7 | 23.5 |
| | MSCC (Song et al., 2024) | 4.40 | 17.4 | 34.7 | 47.9 | 6.87 | 13.1 | 26.3 | 38.9 | 9.79 | 6.8 | 16.3 | 29.0 |
| | ParamNet (Jin et al., 2023) | 2.11 | 24.9 | 53.6 | 71.5 | 3.40 | 16.1 | 38.7 | 58.6 | 6.21 | 9.4 | 22.3 | 39.8 |
| | UVP (Pautrat et al., 2023) | 2.03 | 32.7 | 46.4 | 54.9 | 9.04 | 11.4 | 22.6 | 32.5 | 18.80 | 5.0 | 12.1 | 19.9 |
| | GeoCalib (Veicht et al., 2024) | 0.92 | 53.6 | 73.9 | 82.6 | 2.18 | 28.9 | 52.5 | 69.6 | 5.04 | 12.4 | 28.0 | 45.8 |
| | **Puffin (Ours)** | **0.41** | **78.3** | **91.0** | **95.2** | **0.74** | **60.2** | **81.2** | **90.0** | **1.21** | **42.4** | **70.5** | **84.3** |

incorporating an additional yaw parameter to represent cross-view deviations and conditioning on both the target-view camera map and the VAE-encoded initial view (with text descriptions randomly dropped to support both text-conditioned and text-free generation); and (iii) photographic guidance, where the model suggests camera parameter adjustments from an initial view to achieve images with higher photographic aesthetics. Visualization results are presented in Figure 6.

## 3 EXPERIMENTS

Datasets and benchmarks that span vision, language, and camera modalities remain scarce in the domain of multimodal spatial intelligence. To address this gap, we introduce **Puffin-4M**, a large-scale, high-quality dataset comprising 4 million vision-language-camera triplets. The construction of this curated dataset consists of four stages: panoramic data collection and preprocessing, perspective image generation, scene and spatial reasoning captioning, and extensions for cross-view instruction tuning. Details about the constructed dataset and training recipe of our framework are presented in Appendix A3 and Appendix A4, respectively.

### 3.1 EVALUATIONS ON CAMERA UNDERSTANDING

**Settings.** Following prior works, we compare our method against a range of learning-based camera calibration approaches, including DeepCalib (Lopez et al., 2019), CTRL-C (Lee et al., 2021), MSCC (Song et al., 2024), ParamNet (Jin et al., 2023), and GeoCalib (Veicht et al., 2024), as well as traditional methods such as SVA (Lochman et al., 2021) and UVP (Pautrat et al., 2023). For each image, gravity estimation is evaluated using the angular errors in roll and pitch, while focal length is evaluated through the error in vertical FoV. For all metrics, we report both the median error and the Area Under the Recall Curve (AUC) at thresholds of 1°, 5°, and 10°. We conduct evaluations on three common datasets, MegaDepth (Li & Snavely, 2018), TartanAir (Wang et al., 2020), and LaMAR (Sarlin et al., 2022). Notably, images from these datasets are primarily captured or simulated in well-structured environments, where buildings, rooms, or trees occupy a substantial portion of the scene. Moreover, the camera parameters in some datasets are limited in distribution; for instance,

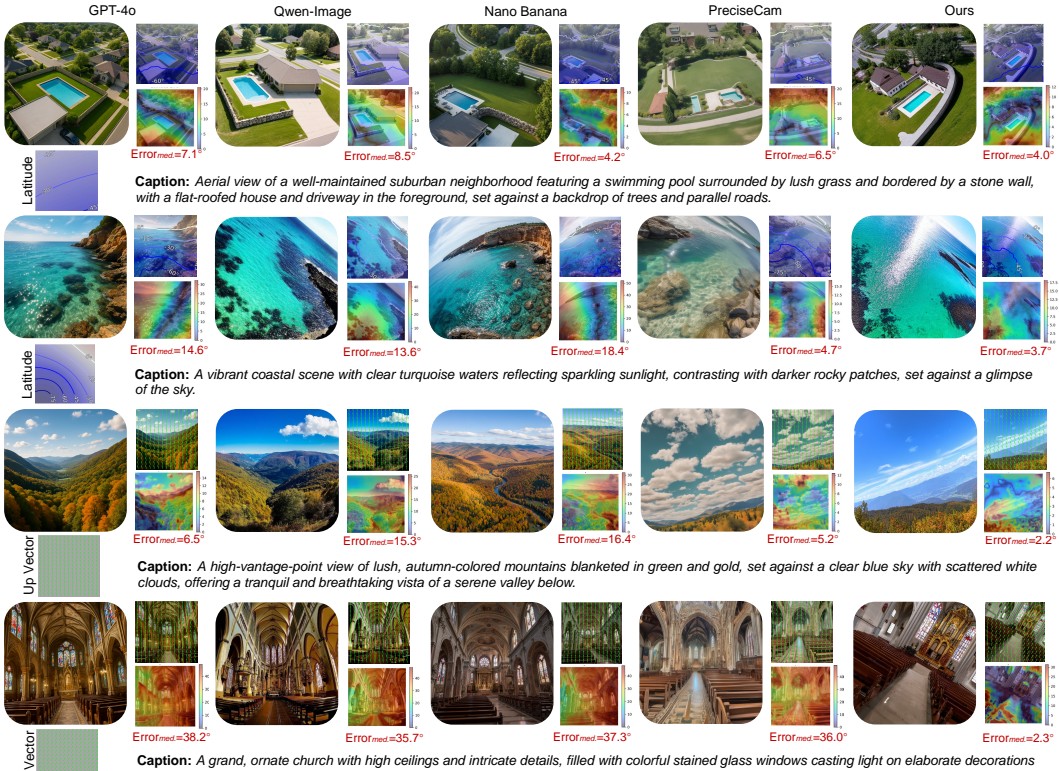

**Figure 4: Comparison results on controllable generation.** We visualize the generated image along with its camera map (latitude or up vector, estimated by (Veicht et al., 2024)), error map to the GT camera map, and the median error. The caption and target camera map are presented at the bottom of each comparison.

**Table 2: Camera-controllable generation evaluation on Puffin-*Gen*.** When evaluating multimodal models, we convert the camera parameters from radians to degrees* or express them using standard photographic terms[†].

| Approach | Up Vector [degrees] | | Latitude [degrees] | | Gravity [degrees] | | Visual Quality |
|---|---|---|---|---|---|---|---|
| | mean error ↓ | median error ↓ | mean error ↓ | median error ↓ | mean error ↓ | median error ↓ | FID ↓ |
| GPT-4o* (OpenAI, 2025) | 24.11 | 22.86 | 15.87 | 13.67 | 28.08 | 27.39 | 95.92 |
| GPT-4o[†] (OpenAI, 2025) | 24.07 | 22.10 | 14.67 | 12.43 | 27.19 | 26.32 | 94.43 |
| Qwen-Image* (Wu et al., 2025a) | 23.80 | 22.73 | 15.76 | 13.90 | 27.75 | 27.22 | 83.31 |
| Qwen-Image[†] (Wu et al., 2025a) | 23.98 | 22.60 | 15.92 | 13.92 | 27.86 | 26.45 | 83.37 |
| Nano Banana* (Google DeepMind, 2025) | 24.08 | 23.13 | 16.66 | 15.05 | 28.78 | 28.22 | 91.66 |
| Nano Banana[†] (Google DeepMind, 2025) | 24.65 | 23.50 | 15.80 | 13.98 | 28.22 | 26.73 | 88.02 |
| PreciseCam (Bernal-Berdun et al., 2025) | 18.66 | 17.47 | 12.49 | 9.99 | 18.39 | 15.34 | 90.91 |
| **Puffin (Ours)** | **11.94** | **10.12** | **6.34** | **4.04** | **6.79** | **3.43** | **69.46** |

TartanAir uses a single FoV for all images. To complement these settings, we construct a more challenging dataset, Puffin-*Und*, designed for a comprehensive assessment of camera understanding. This dataset contains 1,000 images spanning diverse camera configurations and scenarios.

**Comparison Results.** As shown in Table 1, our method outperforms the baselines on MegaDepth and Puffin-*Und*, and achieves comparable results on TartanAir and LaMAR. Due to the fixed image resolution in our training data ($512 \times 512$), we adopt a central cropping strategy followed by resizing to rectangular inputs for evaluating non-square images. The vertical FoV is then computed from the predicted value and scaled according to the crop ratio. Nevertheless, this procedure may discard semantically valid content and thereby degrade our camera understanding performance, particularly when the aspect ratio deviates substantially from unity, as in LaMAR, where Puffin slightly underperforms the state-of-the-art method (Veicht et al., 2024). While this limitation is orthogonal to our current exploration, it could be potentially mitigated in future work by constructing a multi-scale training dataset. We present the horizon lines derived from the predicted camera parameters of different methods in Appendix A5.1 (Figure A9).

**Table 3: Ablation study on camera understanding.** We evaluate our method with different architectures and the mode of thinking with camera.

| Approach | Roll [degrees] | | | | Pitch [degrees] | | | | FoV [degrees] | | | |
|---|---|---|---|---|---|---|---|---|---|---|---|---|
| | error ↓ | AUC ▷ 1/5/10° ↑ | | | error ↓ | AUC ▷ 1/5/10° ↑ | | | error ↓ | AUC ▷ 1/5/10° ↑ | | |
| InternVL3 (Zhu et al., 2025) | 0.91 | 53.7 | 75.5 | 85.6 | 1.72 | 31.9 | 59.7 | 76.3 | 2.96 | 19.7 | 43.1 | 63.5 |
| Qwen2.5-VL (Bai et al., 2025) | 0.79 | 58.8 | 78.0 | 86.5 | 1.61 | 36.4 | 62.4 | 78.0 | 2.91 | 19.4 | 42.5 | 62.5 |
| Vision Encoder (Heinrich et al., 2025) | 0.55 | 69.0 | 86.2 | 92.6 | 1.00 | 49.8 | 74.1 | 85.9 | 1.87 | 28.3 | 57.9 | 75.9 |
| Ours | 0.47 | 75.6 | 89.7 | 94.6 | 0.91 | 54.2 | 77.5 | 87.9 | 1.48 | 38.0 | 66.2 | 81.5 |
| Ours *w/* Thinking | **0.41** | **78.3** | **91.0** | **95.2** | **0.74** | **60.2** | **81.2** | **90.0** | **1.21** | **42.4** | **70.5** | **84.3** |

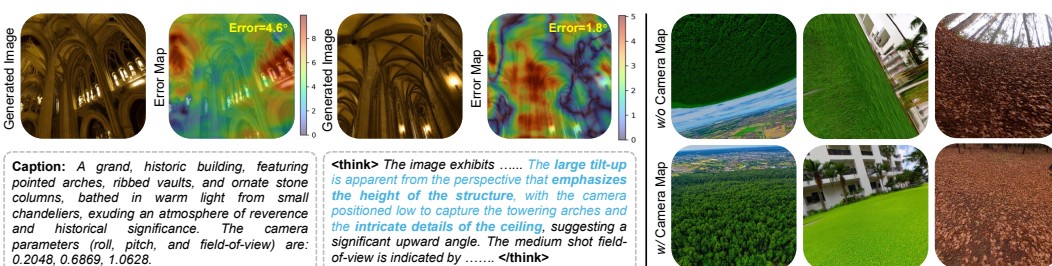

**Figure 5: Ablation study on the camera-controllable generation.** We evaluate the effectiveness of the thinking mode (left) and the precise geometric encoding provided by camera map (right).

## 3.2 EVALUATIONS ON CAMERA-CONTROLLABLE GENERATION

**Settings.** We evaluate our generation performance against the state-of-the-art method Precise-Cam (Bernal-Berdun et al., 2025). In addition, we compare our approach with recent powerful multimodal models, including GPT-4o (OpenAI, 2025), Qwen-Image (Wu et al., 2025a), and Nano Banana (Google DeepMind, 2025), using the same captions as our method. The prompt templates are shown in Appendix A2.3. To mitigate the data gap for the multimodal models, we convert the camera parameters in captions from radians to degrees or express them using professional photographic terms. For quantitative evaluation, we adopt an offline method (Veicht et al., 2024) to estimate pixel-wise camera maps. Using the ground truth maps, we then compute the mean and median errors of the up vector, latitude, and gravity, all measured in degrees. We also incorporate the standard FID metric to assess the overall visual quality of the generated images. Since no benchmark dataset exists for text-to-image generation with precise camera parameters, we construct Puffin-*Gen* to fill this gap. The dataset consists of 650 caption–camera pairs spanning diverse scenarios and camera configurations. The construction details of Puffin-*Gen* and Puffin-*Und* are presented in Appendix A3, and we will release them to support standardized evaluation and facilitate subsequent works.

**Comparison Results.** We report quantitative and qualitative results in Table 2 and Figure 4. Our method outperforms existing multimodal models by a large margin across all metrics. While these models produce high-quality and aesthetically pleasing images, they fail to ensure spatially consistent layouts under specific camera configurations. PreciseCam (Bernal-Berdun et al., 2025) provides effective control but often generates monotonous stylized outputs (*e.g.*, anime) with limited diversity, and struggles with challenging configurations such as significantly tilted poses. In contrast, our method generalizes well across diverse scenarios and camera settings, demonstrating strong practicality for real-world image generation. Additional results and parameter-specific controls are shown in Appendix A5.2 (Figure A11 and Figure A12).

## 3.3 ABLATION STUDIES

**Architecture.** As discussed in Section 2.1, directly fine-tuning the existing VLMs yields a significant performance bottleneck since their vision encoders learn overly condensed high-level features and language models have little prior knowledge of spatial perception. As listed in Table 3, directly finetuning the current VLMs (Bai et al., 2025; Zhu et al., 2025) even underperforms the vision-only network. To this end, we carefully pair an LLM (Qwen et al., 2024) with the vision encoder (Heinrich et al., 2025); both of them are first aligned and then fine-tuned on the camera understanding dataset. By jointly integrating the geometric perception and context understanding in a staged optimization manner, our method (Ours) outperforms the above approaches on all evaluation metrics.

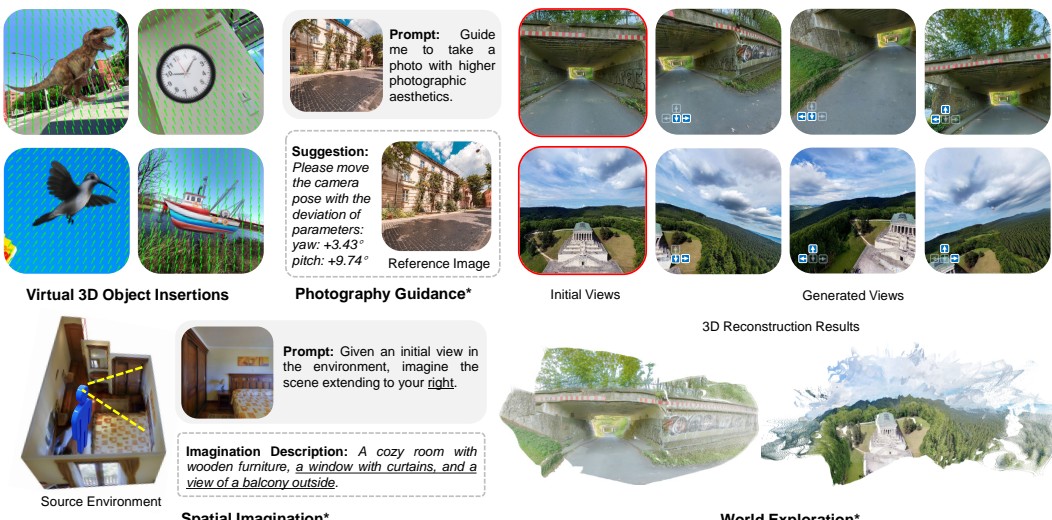

**Figure 6: Applications of the proposed Puffin.** Our method can help 3D object insertion into a wild image by predicting its camera parameters. Additionally, it can flexibly extend to various cross-view tasks such as the spatial imagination, world exploration, and photographic guidance, by instruction tuning*.

**Thinking with Camera.** To mitigate the modality gap between camera and vision–language, we introduce thinking with camera. For camera understanding, we align spatially grounded visual cues with photographic terms across geometric context, enabling LMMs to predict camera parameters through structured spatial reasoning. As shown in Table 3, this design (Ours *w/* Thinking) consistently improves performance, especially for pitch and FoV prediction that depend on broader contextual priors, demonstrating the framework's ability to capture hierarchical spatial context beyond localized geometric cues. Thinking with camera also enhances camera-controllable generation: given a prompt with scene descriptions and target parameters, the model infers potential spatial cues and uses them as a semantic planning stage to guide synthesis. As illustrated in Figure 5 (left), it emphasizes visual cues such as ceilings under a large tilt-up, yielding more accurate spatial simulation.

**Camera Parameters *vs*. Camera Map.** Beyond discrete camera tokens derived from explicit numerical parameters, we further introduce a continuous representation of camera geometry via pixel-wise camera maps for controllable image generation. We show the effectiveness of the precise geometric encoding provided by camera map in Figure 5 (right). Compared to numerical values of the camera parameters, the camera map encodes the local geometric context at each pixel, including orientation and spatial displacement clues, offering precise control over spatial layout and viewpoint. Without the camera map as conditions, generated images may exhibit severe geometric distortions and inverted spatial illusions under challenging camera configurations.

## 3.4 APPLICATIONS

We illustrate the versatile capabilities of our Puffin in Figure 6. Similar to previous methods (Hold-Geoffroy et al., 2018; Jin et al., 2023), Puffin can support virtual 3D object insertion into in-the-wild images by accurately predicting camera parameters. Furthermore, it can be flexibly extended to a range of cross-view tasks through instruction tuning, such as spatial imagination, world exploration, and photographic guidance. For both the initial and generated views in world exploration, we visualize 3D reconstruction results with VGGT (Wang et al., 2025a), showing proper spatial consistency across viewpoints. Additional results are presented in Appendix A5.3 (Figure A13, A14, A15).

## 4 CONCLUSION

We introduce Puffin, a unified multimodal model that jointly performs camera-centric understanding and generation across arbitrary viewpoints. These two tasks have been commonly treated as isolated problems and independently explored by the research community. Yet, in essence, they represent two complementary sides: decoding the geometry of the world and encoding it back into controllable, perceptually consistent visual content. Unlike previous unified models restricted to oversimplified

front-view assumptions, Puffin eliminates the modality gap by interpreting the camera as language and leverages the notion of thinking with camera. We argue that unifying camera-centric understanding and generation anchors perception and synthesis to a shared representation of camera geometry, allowing machines to reason about space more holistically and interactively. Such a unified camera-centric model underpins robust spatial intelligence and fosters more versatile applications.

## ACKNOWLEDGEMENTS

This study is supported under the RIE2020 Industry Alignment Fund Industry Collaboration Projects (IAF-ICP) Funding Initiative, as well as cash and in-kind contribution from the industry partner(s). It is also supported by Singapore MOE AcRF Tier 2 (MOE-T2EP20224-0003). We thank Zhouxia Wang, Zongsheng Yue, Haiwen Diao, Zhaoxi Chen, Zhijie Shen, Xiang Li, Qiqi Gong, Edurne Bernal-Berdun, and Koki Maeda for their insightful discussions. We thank Xu Song and Juncheng Zhou for their help with the comparison methods.

## REPRODUCIBILITY STATEMENT

We have made every effort to ensure reproducibility. The main paper provides a complete description of the framework, and the appendix details the implementation, dataset, and training recipe. We have released the code, models, dataset construction pipeline, and benchmark to further advance research in multimodal spatial intelligence.

## ETHICS STATEMENT

We affirm our adherence to the ICLR Code of Ethics and have conducted this work with integrity and responsibility. Our research does not involve human subjects, sensitive personal data, or applications intended for harmful use. We aim to contribute positively to the community by releasing resources that promote openness, reproducibility, and fair advancement of research.

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

APPENDIX

In this document, we provide the following supplementary content: related work, implementation details, dataset construction, training recipe, additional experiments, limitation and future work, and statements.

## A1 RELATED WORK

### A1.1 LARGE MULTIMODAL MODELS

Built upon a visual encoder (Radford et al., 2021; Zhai et al., 2023; Tschannen et al., 2025) and a large language model (LLM) (Touvron et al., 2023; Qwen et al., 2024; Cai et al., 2024; Liu et al., 2024), LMMs (Liu et al., 2023; Chen et al., 2024; Tong et al., 2024a; Bai et al., 2025; Zhu et al., 2025) process mixed visual and textual inputs and perform understanding and reasoning via language generation. Fueled by large-scale pre-training of the vision and language models and sophisticated instruction-tuning, LMMs excel at high-level understanding tasks, such as object localization, counting, and optical character recognition. However, these models, optimized for semantic alignment between vision and language, remain limited in capturing image intrinsics (*e.g.*, depth and geometry), which constrains their ability in camera understanding and spatial reasoning. To bridge this gap, it is crucial

to enrich LMMs with geometry-aware prior knowledge that preserves structural details beyond semantics. Moreover, aligning such geometric cues with linguistic tokens provides a pathway to extend the reasoning capacity of LMMs from abstract semantics to physically grounded spatial understanding.

### A1.2 UNIFIED MULTIMODAL MODELS

As an extension of standard LMMs, unified multimodal models (Team, 2024; Wang et al., 2024; Tang et al., 2025; Wu et al., 2025d; Lin et al., 2025b; Wu et al., 2024; Tong et al., 2024b; Pan et al., 2025; Lin et al., 2025a; Wu et al., 2025c; Chen et al., 2025; Wu et al., 2025b; Xie et al., 2024; 2025a) jointly learn visual understanding and generation within a single framework. Two main design philosophies are typically adopted. One line of work formulates visual generation as autoregression over either discrete (Team, 2024; Wu et al., 2024; Wang et al., 2024; Wu et al., 2025b) or continuous (Fan et al., 2025) image tokens, sharing LLM parameters for both understanding and generation. Another line (Pan et al., 2025; Chen et al., 2025; Wu et al., 2025c; Lin et al., 2025a) aligns pre-trained LMMs with diffusion modules, enabling faster convergence and lower training cost. While both types of models advance general image understanding and generation, they are constrained to simplistic camera assumptions (*e.g.*, fixed front-view, predefined FoVs), hindering their practical applicability to realistic and complex environments. To this end, we introduce a camera-centric framework that jointly performs camera understanding and controllable generation.

### A1.3 CAMERA GEOMETRY FROM VISION

Tasks such as camera calibration and pose estimation have long been a central topic in 3D vision (Pollefeys et al., 1999; Hartley & Zisserman, 2003; Liao et al., 2023; Veicht et al., 2024; Hold-Geoffroy et al., 2018; Jin et al., 2023; Zhang et al., 2024; Lin et al., 2025c). While earlier learning-based works attempted to directly regress camera parameters from input images (Hold-Geoffroy et al., 2018; Workman et al., 2015; Bogdan et al., 2018; Zhai et al., 2016; Kendall et al., 2015), recent advances increasingly explore the use of intermediate representations or geometry fields to bridge the prediction gap. Representative approaches (Lee et al., 2020; 2021; Song et al., 2024; Janampa & Pattichis, 2024; Yin et al., 2018) leverage geometric structures or semantic features to alleviate the inherent difficulty of inferring camera parameters from a few views. Building on priors of the camera model and the perspective properties of captured images, a growing body of methods proposes to learn dense geometry fields, such as distortion distribution maps (Liao et al., 2020; 2021), pixel displacement fields (Li et al., 2019; Liao et al., 2025; Xie et al., 2025b), camera rays (Zhang et al., 2024), perspective fields (Jin et al., 2023; Veicht et al., 2024; Tirado-Garín & Civera, 2025), or incidence fields (Zhu et al., 2023; He et al., 2025; Deng et al., 2024). However, such representations typically emphasize low-/mid-level patterns, limiting their ability to capture a holistic and coherent spatial concept. Rather than pursuing better representations, this work explores an alternative perspective: interpreting the camera as language.

## A2 IMPLEMENTATION DETAILS

### A2.1 NETWORK CONFIGURATION

For the architecture of our Puffin, we use the pretrained C-RADIOv3-H (Heinrich et al., 2025), Qwen2.5-1.5B-Instruct (Qwen et al., 2024), and SD3-Medium (Esser et al., 2024) to initialize our geometry-aligned vision encoder, LLM, and diffusion model, respectively. Learnable queries with the number of 64 and a lightweight connector comprising six transformer layers are exploited to translate the LLM hidden states to conditioning signals for the diffusion model. The resolutions of the image and camera maps are set to $512 \times 512$ for all tasks. For tokenization, the camera parameter tokenizer follows the same procedure as the text tokenizer. Since camera parameters are numerical, we first serialize them into discrete tokens, which are naturally handled by the standard text tokenizer without requiring any extra module. Introducing an additional tokenizer (or a separate text encoder) would substantially increase the alignment burden across modules and modalities in a unified multimodal model. For this reason, we keep the vanilla decoder-only LLM backbone and its tokenizer to process both language and camera parameters. For the camera map, we adopt Perspective Field (Jin et al., 2023). We first normalize its values to the range $[-1, 1]$, and then reuse the image tokenizer (*i.e.*, the

**Table A1: Camera parameter-to-term mapping.** To align camera parameters (roll, pitch, and FoV) with the prior knowledge space of LMMs, their numerical ranges are mapped to professional photographic terms. Combining different terms linguistically allows us to jointly describe the full camera geometry of an image.

| | Roll | | | | |
|---|---|---|---|---|---|
| **Term** ($t$) | Large counterclockwise Dutch angle | Small counterclockwise Dutch angle | Near level shot | Small clockwise Dutch angle | Large clockwise Dutch angle |
| **Example** |  |  |  |  |  |
| **Parameter** ($p$) | $[-45°, -20°)$ | $[-20°, -5°)$ | $[-5°, 5°]$ | $(5°, 20°]$ | $(20°, 45°]$ |
| | **Pitch** | | | | |
| **Term** ($t$) | Large tilt-down | Small tilt-down | Near straight-on shot | Small tilt-up | Large tilt-up |
| **Example** |  |  |  |  |  |
| **Parameter** ($p$) | $[-45°, -20°)$ | $[-20°, -5°)$ | $[-5°, 5°]$ | $(5°, 20°]$ | $(20°, 45°]$ |
| | **FoV** | | | | |
| **Term** ($t$) | Close-up | Medium shot | Wide-angle | Ultra wide-angle | |
| **Example** |  |  |  |  | |
| **Parameter** ($p$) | $[20°, 35°)$ | $[35°, 65°)$ | $[65°, 90°)$ | $[90°, 105°]$ | |

VAE encoder), as the camera map also has three channels. Pretraining a specialized tokenizer for camera maps is left as future work.

## A2.2 CAMERA PARAMETERS TO PHOTOGRAPHIC TERMS

To bridge the gap between the detailed numerical values of camera parameters and the highly abstracted understanding capability learned by LMMs, we propose using professional photographic terms as intermediate supervision for our framework. Specifically, we quantize the range of each camera parameter and map them to the following photographic terms: (i) Roll: large counterclockwise Dutch angle, small counterclockwise Dutch angle, near level shot, small clockwise Dutch angle, large clockwise Dutch angle; (ii) Pitch: large tilt-down, small tilt-down, near straight-on shot, small tilt-up, large tilt-up; (iii) FoV: close-up, medium shot, wide-angle, ultra wide-angle. The detailed mapping relationship between the camera parameters and photographic terms is listed in Table A1.

## A2.3 PROMPT TEMPLATE FOR MULTIMODAL TASKS

For text-to-image controllable generation, we use the following prompt template to format user instructions: `User: Generate an image: <caption>\n Assistant:`. The `<caption>` includes both the image description and the numerical camera parameters (roll, pitch, FoV, all in radians). For camera understanding, we employ the following prompt template to format the basic user instruction: `User: <image><question>\n Assistant:`. The `<question>` can be set as "`Describe the image in detail. Then reason its spatial distribution and estimate its camera parameters (roll, pitch, and field-of-view)`". For cross-view camera-controllable generation, the prompt template is formatted as: "`User: Generate a target image given an initial view: <image><caption>\n Assistant:`". Here, `<image>` denotes the initial view token from the image tokenizer, while `<caption>` represents the target image description along with the target camera parameters (roll, pitch, yaw, and FoV, all in radians). During cross-view instruction tuning, we randomly set the `<caption>` to null with a probability of 0.5, thereby enabling both text-free and text-conditioned image-to-image generation. When applying the spatial reasoning paradigm, we switch to a new `<question>` for camera understanding: "`Reason the spatial distribution of this image in a thinking mode, and then estimate its camera parameters (roll, pitch, and field-of-view)`". For generation, we first enrich the vanilla prompt using our model with the template: "`User: <caption><question>\n Assistant:`". Here, `<caption>` refers to the vanilla image description, and `<question>` is "`Given a scene description and corresponding camera parameters, merge them into a coherent prompt and generate an accurate visualization that highlights visual cues for spatial reasoning`". For other instruction tuning tasks, `<question>` is set to "`Given

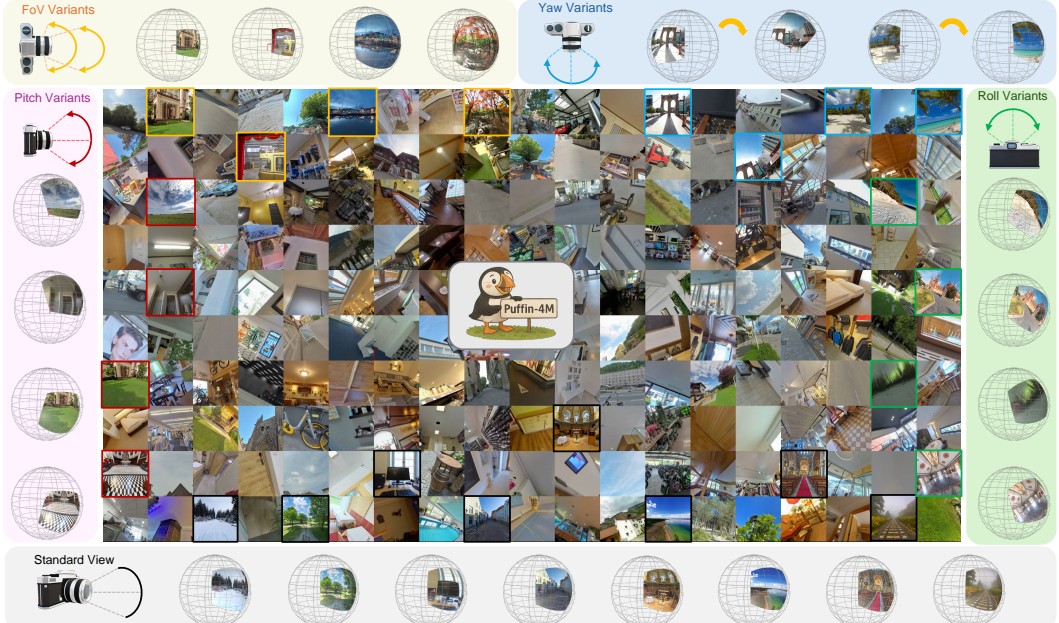

**Figure A1: Overview of the proposed Puffin-4M dataset.** It consists of 4 million vision-language-camera triplets under various scenarios and camera configurations. We mark the sample images with different colors, each denoting a different variant of the camera configurations.

```
the initial view and the camera parameters of the target view with
the deviation yaw angle, how would you describe the target image
to build a replica of the scene?"
```
for spatial imagination, and "Estimate the
```
camera parameters (roll, pitch, and field-of-view) of this image.
And then predict the deviation camera yaw angle and pitch angle
of the target view with high photographic aesthetics."
```
for photographic guidance.

## A3 DATASET CONSTRUCTION

The overview of our Puffin-4M is shown in Figure A1. The construction of this curated dataset consists of four stages: panoramic data collection and preprocessing, perspective image generation, scene and spatial reasoning captioning, and extensions for cross-view scenarios. Following previous works (Veicht et al., 2024; Jin et al., 2023; Bernal-Berdun et al., 2025; Hold-Geoffroy et al., 2018), we render the perspective images with various camera intrinsic and extrinsic parameters from 360° panoramic images using a standard camera model. The pipeline of the dataset construction is illustrated in Figure A2 and the dataset comparison with previous works is listed in Table A2. Puffin-*Und* and Puffin-*Gen* are constructed by exactly the same pipeline as Puffin-4M; the only difference lies in the source images and the task-specific splits. More details are described as follows.

### A3.1 PANORAMIC DATA COLLECTION AND PREPROCESSING

We begin by collecting panoramic images from publicly available datasets (Bertel et al., 2020; Choi et al., 2023; Cao et al., 2025; Huang et al., 2024; Xu et al., 2022; Zhang et al., 2020; Bernal-Berdun et al., 2023; Cheng et al., 2018; Bolduc et al., 2023; Chang et al., 2018; Veicht et al., 2024; Armeni et al., 2017) as well as from online platforms (Fli; Str; Wik; HDR; Pol; Amb; Ble; You). In addition, we acquire a large volume of outdoor panoramic data from Google Street View (Goo), spanning 12 cities across Asia, Europe, and North America. In total, our curated dataset comprises approximately 200K high-quality panoramic images with substantial diversity. A significant portion of these images reaches 4K resolution or higher, up to 10K. However, due to variations in 360° camera calibration and acquisition stability, some panoramas exhibit geometric distortions and misalignment. To mitigate this, we apply geometric correction techniques based on line segmentation and vanishing point

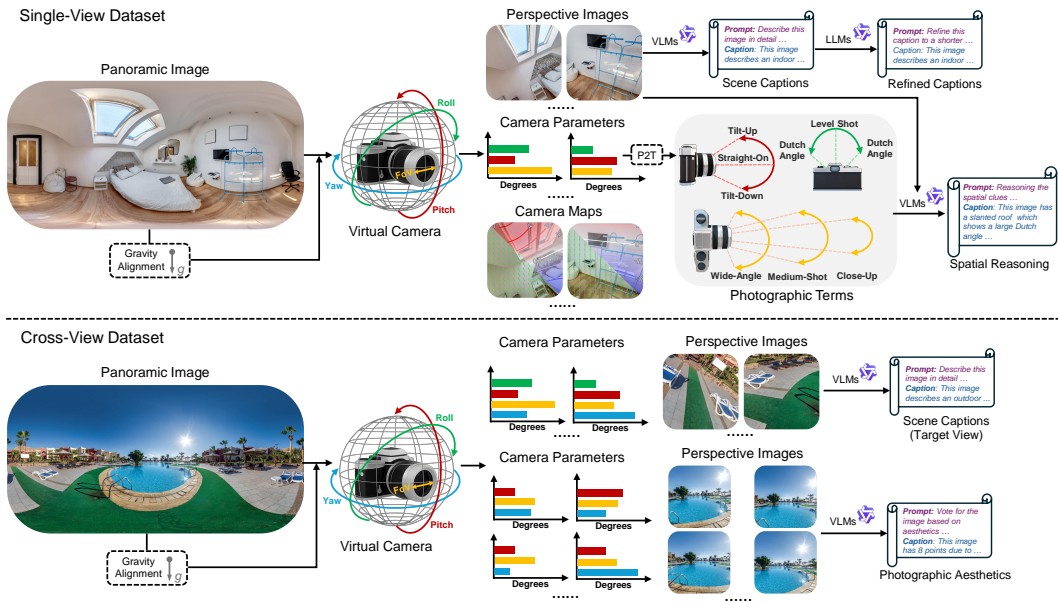

**Figure A2: Pipeline of the dataset construction.** P2T denotes the mapping from the numerical camera parameters to the professional photographic terms. For clarity, we omit the orientations "clockwise" and "counterclockwise" of the Dutch angle in photographic terms.

**Table A2: Dataset Comparisons.** The datasets proposed in previous individual models tailored for camera understanding (Lee et al., 2021; Bogdan et al., 2018; Veicht et al., 2024; Jin et al., 2023; Hold-Geoffroy et al., 2018) or camera-controllable image generation (Bernal-Berdun et al., 2025) *vs.* our Puffin-4M for the camera-centric unified multimodal model. In addition to its larger scale, our dataset also offers advantages in spatial reasoning captions, and cross-view image pairs. For the camera parameters, we denote the intrinsic parameters: focal length ($f$), radial distortion coefficient ($\xi$); and the extrinsic parameters: roll ($\phi$), pitch ($\theta$), yaw ($\psi$).

| Dataset | Task Type | Intrinsics | Extrinsics | # Frames | Details | | | | |
|---|---|---|---|---|---|---|---|---|---|
| | | | | | Camera | Text | Reasoning | Single-View | Cross-View |
| GeoCalib (Veicht et al., 2024) | Understanding | $\{f, \xi\}$ | $\{\phi, \theta\}$ | 37K | ✔ | ✗ | ✗ | ✔ | ✗ |
| CTRL-C (Lee et al., 2021) | Understanding | $f$ | $\{\phi, \theta\}$ | 45K | ✔ | ✗ | ✗ | ✔ | ✗ |
| Deepcalib (Bogdan et al., 2018) | Understanding | $\{f, \xi\}$ | - | 67K | ✔ | ✗ | ✗ | ✔ | ✗ |
| ParamNet (Jin et al., 2023) | Understanding | $f$ | $\{\phi, \theta\}$ | 190K | ✔ | ✗ | ✗ | ✔ | ✗ |
| Perceptual (Hold-Geoffroy et al., 2018) | Understanding | $f$ | $\{\phi, \theta\}$ | 390K | ✔ | ✗ | ✗ | ✔ | ✗ |
| PreciseCam (Bernal-Berdun et al., 2025) | Generation | $\{f, \xi\}$ | $\{\phi, \theta\}$ | 57K | ✔ | ✔ | ✗ | ✔ | ✗ |
| **Puffin-4M (Ours)** | Unified | $f$ | $\{\phi, \theta, \psi\}$ | 4M | ✔ | ✔ | ✔ | ✔ | ✔ |

estimation (Jiang et al., 2022; Zou et al., 2018), aligning the panoramas with the gravity direction and improving structural consistency.

## A3.2 PERSPECTIVE IMAGE GENERATION

We adopt the pinhole camera model with varying intrinsic parameters (vertical FoV) and extrinsic parameters (roll and pitch) to synthesize perspective images, following the protocol established in recent state-of-the-art camera calibration works (Veicht et al., 2024; Jin et al., 2023). For each panoramic image, we generate multiple perspective crops by uniformly sampling roll, pitch, and vertical FoV within the ranges $[-45°, 45°]$, $[-45°, 45°]$, and $[20°, 105°]$, respectively. The number of crops is adaptively determined based on the resolution of the original panorama. While our current setup assumes an ideal pinhole model, incorporating radial distortion effects via an additional distortion parameter $\mathbf{k}$ is left as future work. After generating the perspective images, we further convert the corresponding camera parameters into a pixel-wise Perspective Field (Jin et al., 2023) as camera map, where each pixel is annotated with its up-vector $\mathbf{u_x}$ and latitude angle $\varphi_\mathbf{x}$ to enable fine-grained spatial encoding:

$$\mathbf{u_x} = \lim_{c \to 0} \frac{\mathcal{P}(\mathbf{X} - c\mathbf{g}) - \mathcal{P}(\mathbf{X})}{\|(\mathcal{P}(\mathbf{X}) - c\mathbf{g}) - \mathcal{P}(\mathbf{X})\|_2}, \ \varphi_\mathbf{x} = \arcsin\left(\frac{\mathbf{R} \cdot \mathbf{g}}{\|\mathbf{R}\|_2}\right), \quad (1)$$

where $\mathcal{P}(\mathbf{X}) = \mathbf{x}$ denotes the mapping of a 3D point $\mathbf{X}$ to an image pixel $\mathbf{x}$, and each pixel $\mathbf{x}$ corresponds to a light ray $\mathbf{R}$ originating from $\mathbf{X}$.

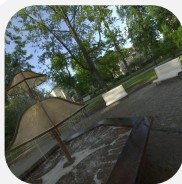

###[System Role Instruction]
Given that the capturing camera information for this image is known—horizontal orientation: large clockwise Dutch angle; vertical orientation: near straight-on; field-of-view: wide-angle—analyze the visual-spatial cues to estimate these three parameters. Integrate both Observation and Reasoning into a single coherent explanation without redundancy. Your output must consist of exactly one paragraph enclosed within <think> and </think> tags. The explanation should reflect visually grounded reasoning using rich spatial cues from both local semantics and global scene layout. Structure the response into three concise sentences, each contributing to the overall inference, without explicitly labeling individual camera parameters or describing the general scene. Do not include any extra information or format deviations.

<think> The image exhibits a large clockwise Dutch angle, as evidenced by the tilted perspective of the scene, where the ground and horizontal elements such as the bench and pathway are not aligned with the frame's edges. The vertical orientation appears to be near straight-on, as the vertical lines of the trees and structures maintain a consistent alignment with the frame's vertical axis, suggesting minimal tilt in that direction. The wide-angle field-of-view is apparent from the expansive view of the park, capturing a broad range of elements including the pathway, benches, trees, and distant buildings, which are all visible within a single frame, indicating a broad spatial coverage. </think>

**(a)** The prompt to generate the reasoning caption for thinking with camera.

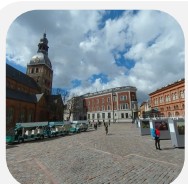

###[System Role Instruction]
You are an expert photography judge. For each input image, where only camera yaw and pitch vary, evaluate its aesthetic quality using the following 4 criteria (total 10 points):
1. Viewpoint Creativity (0–2 points): Is the perspective (yaw/pitch) expressive or unique? Does it offer a refreshing view of the subject or scene?
2. Subject Emphasis (0–2 points): How well does the pose highlight and isolate the main subject? Is the framing intentional and focused? Is the main subject complete and in visual focus, such as at the point of thirds or the golden section?
3. Compositional Balance (0–2 points): Is the image visually stable or artfully tilted? Is horizon placement and element alignment coherent?
4. Spatial Harmonization (0–4 points): Does the scene exhibit harmonized spatial distribution? Are the foreground, midground, and background arranged to convey depth or balance? Does the space feel coherent and deliberate?

The image depicts a public square with a mix of architectural elements, including a prominent church with a tall spire, a modern building, and a red-brick building. There is a small train-like vehicle in the foreground, and a few people are scattered across the scene. The sky is partly cloudy, adding a dynamic element to the composition. Below is the evaluation based on the specified criteria:

#### **1. Viewpoint Creativity (Score: 1/2): **
- The perspective appears to be a standard eye-level view, which is common for capturing urban scenes. There is no particularly unique or expressive yaw or pitch angle that offers a refreshing or unconventional viewpoint.
- The image is straightforward and functional, focusing on capturing the scene rather than experimenting with perspective.
#### **2. Subject Emphasis (Score: 1/2):**
- The main subject appears to be the church with the tall spire, which is a significant architectural feature in the background. However, the subject is not the sole focus; the image also includes the train, people, and other buildings, which share the visual attention.
- The framing is somewhat balanced but not tightly focused on a single subject. The church is positioned near the left side, adhering somewhat to the rule of thirds, but it is not the only point of interest.
#### **3. Compositional Balance (Score: 2/2):**
- The horizon is relatively level, and the elements in the image are aligned coherently. The buildings and the train are parallel to the ground, contributing to a sense of stability.
- The image is visually balanced, with the church on the left, the modern building in the center, and the red-brick building on the right. The distribution of elements feels intentional and organized.
#### **4. Spatial Harmonization (Score: 4/4):**
- The image effectively conveys depth through the arrangement of the foreground, midground, and background. The cobblestone pavement in the foreground, the train and people in the midground, and the buildings in the background create a layered composition.
- The spatial distribution is harmonious, with each element contributing to the overall balance. The church spire draws the eye upward, adding vertical interest, while the horizontal lines of the buildings and train provide a sense of order.

**Total Score: 8/10**

**(b)** The prompt to generate the photographic aesthetic score for the photographic guidance task.

**Figure A3: Examples of the designed prompts for captioning our dataset.**

### A3.3 SCENE AND SPATIAL REASONING CAPTIONS

Captions are essential for multimodal understanding and generation. To construct high-quality descriptions, we first utilize Qwen2.5-VL-7B-Instruct (Bai et al., 2025) to generate semantically rich captions for each perspective image. These are subsequently distilled using Qwen2.5-7B-Instruct (Qwen et al., 2024) into shorter, vivid, and visually expressive sentences.

To bridge the modality gap between camera geometry and vision-language representations, we introduce the notion of thinking with camera that explicitly guides multimodal tasks using spatially grounded visual cues and professional photographic terms. For captioning such a spatial reasoning process, we propose a two-step strategy: first, we map the numerical camera parameters to their corresponding semantic photographic terms (see Table A1); then, for each camera parameter, we use its corresponding photographic terms to *retrieve* and prompt out relevant visual concepts from the LMM's prior knowledge. For the trade-off between accuracy and efficacy, we employ Qwen2.5-VL-32B-Instruct to generate the final spatial reasoning captions.

### A3.4 CROSS-VIEW DATASET

In addition to the single-view dataset construction described above, we further enrich our dataset with cross-view data involving various tasks. Specifically, during cross-view perspective image generation, we extend the camera's degrees of freedom by incorporating the yaw angle in addition to roll and pitch, sampling it uniformly from $[0°, 360°)$. The initial view is initialized with a standard camera pose (roll = pitch = yaw = $0°$), and the target view is rendered using a random camera pose sampled within the aforementioned ranges. The target view in each cross-view pair is then captioned using Qwen2.5-VL-7B-Instruct.

**Table A3: Training recipe of Puffin.** For the data sampling ratio, we mark the data involving the spatial reasoning and instruction tuning in light blue and light red , respectively. For clarity, we abbreviate the generation and understanding as *Gen.* and *Und.*.

| | Stage I | Stage II | Stage III | Stage IV |
|---|---|---|---|---|
| **Hyperparameters** | | | | |
| Learning rate | $1 \times 10^{-4}$ | $2 \times 10^{-5}$ | $1 \times 10^{-5}$ | $5 \times 10^{-6}$ |
| LR Scheduler | | Cosine | | |
| Weight Decay | | 0.05 | | |
| Betas | | (0.9, 0.95) | | |
| Optimizer | | AdamW | | |
| Batch Size | 1024 | 1024 | 512 | 256 |
| Training Steps | 10K | 30K | 60K | 20K |
| Warm-up Steps | 1K | 900 | 1.8K | 600 |
| LLM | Frozen | Trainable | Trainable | Trainable |
| Diffusion Model | Frozen | Trainable | Trainable | Trainable |
| Vision Encoder | Frozen | Trainable | Trainable | Frozen |
| **Data Sampling Ratio** | | | | |
| Text-Camera→Image (*Gen.*) | 0.5 | 0.5 | 0.0 | 0.0 |
| Image→Text-Camera (*Und.*) | 0.5 | 0.5 | 0.0 | 0.0 |
| Text→Text | 0.0 | 0.0 | 0.33 | 0.0 |
| Text-Camera→Image (*Gen.*) | 0.0 | 0.0 | 0.33 | 0.0 |
| Image→Text-Camera (*Und.*) | 0.0 | 0.0 | 0.33 | 0.0 |
| Image-Camera→Text (Cross-view *Und.*) | 0.0 | 0.0 | 0.0 | 0.47 |
| Image-(Text)-Camera→Image (Cross-view *Gen.*) | 0.0 | 0.0 | 0.0 | 0.47 |
| Image→Camera (Photography *Und.*) | 0.0 | 0.0 | 0.0 | 0.06 |

For the photographic guidance task, we first consulted photography experts and enthusiasts to identify four key aspects of photographic aesthetics: viewpoint creativity, subject emphasis, compositional balance, and spatial harmonization. These are then formulated into four criteria that serve as aesthetic rating prompts for LMMs. We focus on pitch and yaw as the key controllable camera parameters[2]. An initial view is generated with a random pitch in $[-20°, 20°]$, and $N$ neighboring views are sampled by perturbing pitch and yaw within the same range. All rendered views are evaluated by Qwen2.5-VL-32B-Instruct using the aesthetic prompts, and scores are assigned through voting. The pitch and yaw offsets between the initial view and the highest-scoring view are taken as labels for the photographic guidance task.

### A3.5 PROMPT DESIGN

For the scene caption of each image, the prompt is formatted as: "`Describe this image in 3-4 sentences`". Then, we refine the caption into a more compact description using: "`Here is a detailed image description: <caption>. Rewrite it into a much shorter, vivid, and visually rich sentence (one or two sentences) that captures only the most essential elements and atmosphere of the scene. Ensure the description is concise, clear, and optimized for use with a text-to-image generation model`". For captioning the spatial reasoning and photographic aesthetics of each image, we show the corresponding prompts and example results in Figure A3.

## A4 TRAINING RECIPE

The whole training process takes around 4 days with 64 NVIDIA A100 (80 GB) GPUs. In reference, we use greedy search for text generation in camera understanding and set the CFG weight as 4.5 for the camera-controllable image generation. We conduct a multi-stage training strategy, where the vision encoder, LLM, and the diffusion model are aligned in the first stage. Then, in the supervised fine-tuning (SFT) stage, the models are jointly optimized using both base and thinking datasets. Finally, an instruction-tuning stage is applied, involving various cross-view generation and understanding tasks. We elaborate each training stage as follows.

- **Stage I-Alignment**. In this stage, we align the vision encoder with the LLM by training only the MLP projector for the understanding task, where the framework learns to predict

---

[2]Both professional and amateur users generally prefer near level-shot photography, as humans are highly sensitive to horizontal perturbations (Howard & Templeton, 1966; Dyde et al., 2006). Thus, we fix roll at $0°$ for this task.

Table A4: Additional evaluation results on camera understanding.

| Approach | Roll [degrees] | | | | Pitch [degrees] | | | | FoV [degrees] | | | |
|---|---|---|---|---|---|---|---|---|---|---|---|---|
| | error ↓ | AUC ▷ 1/5/10° ↑ | | | error ↓ | AUC ▷ 1/5/10° ↑ | | | error ↓ | AUC ▷ 1/5/10° ↑ | | |
| DeepCalib (Lopez et al., 2019) | 1.59 | 33.8 | 63.9 | 79.2 | 2.58 | 21.6 | 46.9 | 65.7 | 6.67 | 8.1 | 20.6 | 37.6 |
| Perceptual (Hold-Geoffroy et al., 2018) | 2.08 | 26.8 | 53.8 | 70.7 | 3.17 | 21.5 | 41.8 | 57.8 | 13.84 | 2.8 | 7.7 | 16.1 |
| CTRL-C (Lee et al., 2021) | 3.04 | 23.2 | 43.0 | 56.9 | 3.43 | 18.3 | 38.6 | 53.8 | 8.50 | 7.7 | 18.2 | 31.5 |
| MSCC (Song et al., 2024) | 3.43 | 13.5 | 36.8 | 57.3 | 2.64 | 22.6 | 45.0 | 60.5 | 5.81 | 9.6 | 23.8 | 41.6 |
| ParamNet (Jin et al., 2023) | 1.14 | 44.6 | 73.9 | 84.8 | 1.94 | 29.2 | 56.7 | 73.1 | 9.01 | 5.8 | 14.3 | 27.8 |
| SVA (Lochman et al., 2021) | - | 21.7 | 24.6 | 25.8 | - | 15.4 | 19.9 | 22.4 | - | 6.2 | 11.5 | 15.2 |
| UVP (Pautrat et al., 2023) | 0.52 | 65.3 | 74.6 | 79.1 | 0.95 | 51.2 | 63.0 | 69.2 | 3.65 | 22.2 | 39.5 | 51.3 |
| GeoCalib (Veicht et al., 2024) | 0.40 | 83.1 | 91.8 | 94.8 | 0.93 | 52.3 | 74.8 | 84.6 | 3.21 | 17.4 | 40.0 | 59.4 |
| **Puffin (Ours)** | **0.26** | **96.6** | **99.0** | **99.4** | **0.48** | **82.0** | **93.6** | **96.7** | **2.30** | **23.4** | **51.2** | **71.4** |

*(Row group label: Stanford2D3D)*

both text descriptions and camera parameters from the input image. For generation, the framework takes text descriptions, camera parameters, and the camera map as inputs, and learns to synthesize the target image with the corresponding description and configuration. Specifically, we train learnable queries and a connector to bridge the LLM and the diffusion transformer, where the connector maps LLM hidden states into conditioning signals for the diffusion model. A cross-entropy loss and diffusion loss supervise the understanding and generation, respectively, while parameters of the vision encoder, LLM, and diffusion model remain frozen.

- **Stage II-SFT.** After aligning different modalities, we unfreeze all modules except the VAE and fine-tune the entire framework, using the same inputs and outputs as in Stage I. To stabilize training, we apply gradient scaling of 0.1 to the vision encoder.

- **Stage III-SFT *w/ Thinking*.** To further bridge the modality gap between the camera and vision-language, we introduce *thinking with camera* in this stage. The implementation is the same as Stage-II, except that the training data contains spatial reasoning captions (the details of obtaining such captions are provided in Section A3). Beyond generation and understanding, this stage also learns the textual reasoning task, which enriches the vanilla captions with spatially grounded visual cues and translates specific camera parameter values into professional photographic terms.

- **Stage IV-Instruction Tuning.** Finally, we improve our model's ability to adapt to diverse spatial configurations. In particular, three types of cross-view data are trained simultaneously, including the spatial imagination, world exploration, and photographic guidance. The KV cache mechanism is utilized in cross-view generation. The vision encoder is frozen while other modules are trainable.

We release three model variants: Puffin-Base, Puffin-Thinking, and Puffin-Instruct, to accommodate different application needs. Puffin-Base provides a foundation model for unified camera understanding and camera-controllable image generation; Puffin-Thinking enhances spatial reasoning and generation; and Puffin-Instruct is optimized by instruction tuning, supporting cross-view tasks and complex multimodal interactions.

## A5 ADDITIONAL EXPERIMENTS

### A5.1 CAMERA UNDERSTANDING

**Results.** Note that our training dataset consists of images rendered from the source panoramas in Stanford2D3D (Armeni et al., 2017). Although the sampled perspective images and camera parameters differ from those in the test set (Armeni et al., 2017), we exclude these results from the main evaluation to ensure rigor and report them in Table A4 only for reference.

We show more visualization results on the proposed thinking with camera for camera understanding in Figure A8. Qualitative evaluations of the camera understanding methods with horizon line visualization are illustrated in Figure A9. Compared to prior approaches, our Puffin demonstrates strong performance not only in common scenarios such as architectural and indoor scenes, but also in challenging cases characterized by limited geometric features or significantly tilted camera poses. These results highlight the robustness of the proposed method. We visualize additional camera understanding results (with camera maps converted from the predicted camera parameters) on diverse inputs, including AIGC images (OpenAI, 2025) and real-world photographs, in Figure A10.

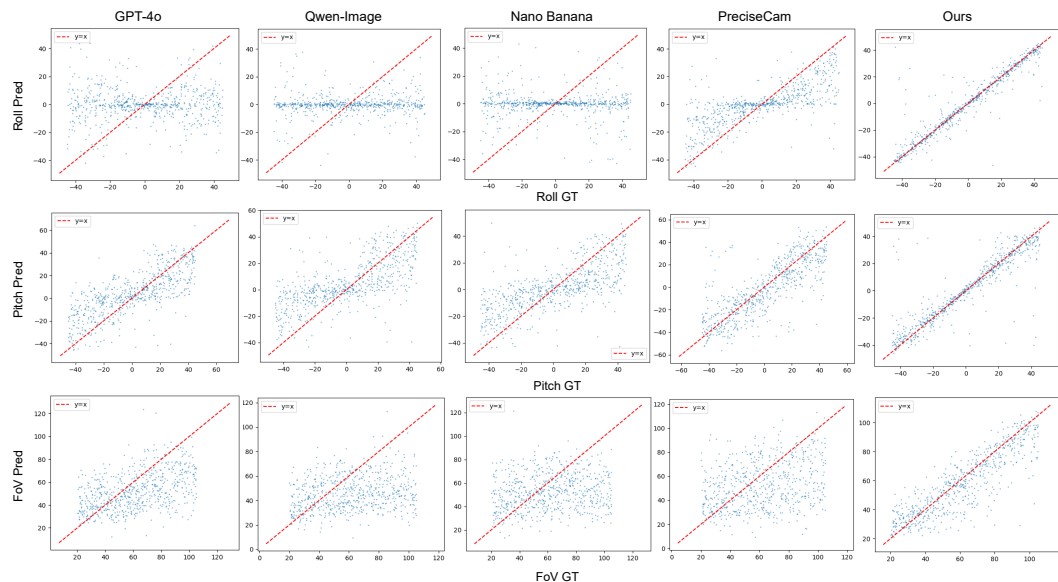

**Figure A4: The predicted *vs.* ground truth camera parameters across all generated samples.** Compared with previous methods, our generated results well align with the distribution of the ground truth camera parameters.

**Discussion.** From the evaluation of existing methods (Table 1), we observe that estimating pitch and FoV is considerably more challenging than estimating roll. This difference arises from the nature of the underlying visual cues. Roll estimation is supported by mid-level geometric representations, such as edges and vanishing lines, which are directly embedded in the image structure and thus relatively straightforward to learn. In contrast, accurate estimation of pitch and FoV requires more extensive contextual priors. Unlike previous vision-based approaches, our method explicitly models the relationship between physical camera parameters and spatial context using a large multimodal model. This integration allows the model to capture spatially contextual knowledge that cannot be sufficiently represented through visual features alone.

### A5.2 CAMERA-CONTROLLABLE GENERATION

**Results.** Our camera-controllable generation results with various camera configurations are shown in Figure A11, and the text-to-image generation (single-view) results with specific controls for each camera parameter are presented in Figure A12.

**Discussion.** We further conduct an in-depth analysis to understand why existing image generation models fail to achieve accurate spatial simulation. Specifically, we decouple the spatial distributions of the generated images with respect to three camera parameters: roll, pitch, and FoV. As illustrated in Figure A4, we visualize scatter plots of the predicted *vs.* ground truth camera parameters (with the reference line $y = x$) across all generated samples. For fairness, the predicted camera parameters are obtained using the state-of-the-art vision-based camera calibration method (Veicht et al., 2024).

Interestingly, we observe a reversed role of camera parameters in controllable generation compared with camera understanding. Specifically, images generated by previous methods (OpenAI, 2025; Wu et al., 2025a; Google DeepMind, 2025; Bernal-Berdun et al., 2025) exhibit poor simulation accuracy on roll compared to pitch, where the predicted roll values fail to align with the target ground truth. In contrast, roll estimation in camera understanding is generally easier than pitch, due to its explicit link with geometric structures.

We attribute this discrepancy to two main factors: (i) Most existing image generation models are trained on datasets curated for high visual aesthetics. Both professional and amateur photographers tend to prefer near-level shots, as humans are sensitive to horizontal perturbations (He et al., 2013). Consequently, variations in roll often conflict with aesthetic preferences, leading to a skewed dataset distribution with far fewer roll variants compared to pitch or FoV. (ii) Roll directly alters the perceived gravity direction in an image, thereby reformulating the common sense of spatial layout. For instance, a strong Dutch angle can make the sea surface appear above the horizon line, creating an inverted

**Table A5: Model size comparisons:** the specialized understanding and generation models (GeoCalib (Veicht et al., 2024) and PreciseCam (Bernal-Berdun et al., 2025)), and the proposed unified camera-centric model.

| Model | Type | Parameters | GFLOPs |
|---|---|---|---|
| GeoCalib (Veicht et al., 2024) | Specialized Model (Understanding) | 28.9M | $3.47 \times 10^2$ |
| PreciseCam (Bernal-Berdun et al., 2025) | Specialized Model (Generation) | 1.3B | $2.67 \times 10^3$ |
| **Puffin-4M (Ours)** | Unified Model | 4.4B | $2.92 \times 10^5$ |

spatial illusion. Such cases are inherently more difficult to simulate, whereas pitch and FoV changes typically only affect the viewing scope without fundamentally disrupting the physical law.

### A5.3 DOWNSTREAM APPLICATIONS

We visualize more downstream application results by instruction tuning here. Specifically, image-to-image generation (cross-view) results with varying yaw angles are shown in Figure A13. World exploration results are provided in Figure A14. Examples of the spatial imagination and photographic guidance are shown in Figure A15.

### A5.4 ABLATION STUDY

We provide additional results and analyses of the ablation study in this part, especially for the mutual effect between camera-centric understanding and generation.

**Single Task *vs*. Unified Training.** In addition to performing multimodal tasks within a unified framework, we aim to exploit the mutual benefits between understanding and generation through joint training. Unlike previous works (Pan et al., 2025; Wu et al., 2025c), we jointly optimize both the LLM and the diffusion model across understanding and generation tasks. This strategy avoids the representational bottleneck imposed by frozen modules and fosters a bidirectional synergy between the two tasks. As illustrated in Figure A5(a)(c), training the camera understanding component in isolation underperforms compared to the unified framework, as the generation process contributes auxiliary diffusion loss at the low-level appearance, which implicitly enhances detailed geometric perception. While the performance gain for generation is less pronounced than for under-

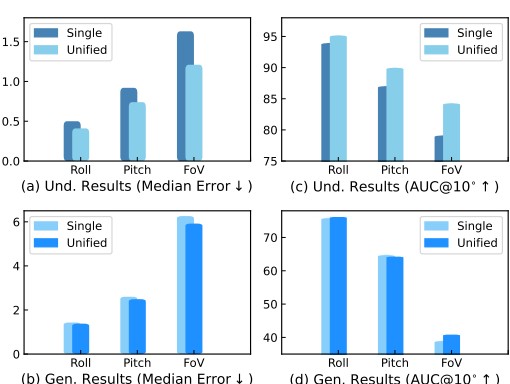

**Figure A5: Ablation study on the mutual effect** between camera understanding (a)(c) and camera-controllable generation (b)(d) supervision.

standing in Figure A5(b)(d), notable improvements emerge in challenging scenarios such as FoV simulation, which requires prior knowledge regarding precise and holistic spatial understanding within an image.

For other general-purpose unified models, the understanding tasks mainly target high-level concepts such as recognition and semantic comprehension. As a result, the domain gaps across multimodal tasks are more pronounced in these models, likely requiring more delicate architectural designs to harmonize representations across different modalities.

### A5.5 ANALYSIS

**Model Size.** We show the comparison results on the total parameter count and FLOPs with previous understanding and generation models in Table A5, such as GeoCalib Veicht et al. (2024) and PreciseCam Jin et al. (2023). While Puffin is larger than previous specialized models, it replaces separate understanding and generation networks with a single unified model that handles both tasks within one framework. This design not only simplifies deployment, but also allows us to fully exploit a high-capacity backbone when training on large-scale multimodal datasets. In terms of overall parameter count and FLOPs, Puffin remains substantially more affordable than recent general-purpose unified multimodal models such as Bagel (14B) Deng et al. (2025) and Qwen-Image (20B) Wu et al. (2025a).

**Table A6: Additional evaluation results on camera understanding** by re-training the comparison method (Veicht et al., 2024) on our constructed Puffin-4M dataset*.

| Approach | Roll [degrees] | | | | Pitch [degrees] | | | | FoV [degrees] | | | |
|---|---|---|---|---|---|---|---|---|---|---|---|---|
| | error ↓ | AUC ▷ 1/5/10° ↑ | | | error ↓ | AUC ▷ 1/5/10° ↑ | | | error ↓ | AUC ▷ 1/5/10° ↑ | | |
| GeoCalib* (Veicht et al., 2024) | 1.12 | 46.0 | 69.3 | 80.1 | 2.54 | 24.5 | 47.5 | 65.4 | 5.47 | 10.8 | 25.5 | 43.8 |
| GeoCalib (Veicht et al., 2024) | 0.92 | 53.6 | 73.9 | 82.6 | 2.18 | 28.9 | 52.5 | 69.6 | 5.04 | 12.4 | 28.0 | 45.8 |
| **Puffin (Ours)** | **0.41** | **78.3** | **91.0** | **95.2** | **0.74** | **60.2** | **81.2** | **90.0** | **1.21** | **42.4** | **70.5** | **84.3** |

**Error Analysis.** In addition to the metrics (error and AUC) used in the main experiments, we also add the error bars for the representative baselines and our method in Figure A6(a) to provide a clearer and more comprehensive comparison. The results show that Puffin exhibits better robustness across the entire data distribution. The improvement is consistent across all intrinsics/extrinsics components and remains stable even under challenging camera configurations.

**Model *vs*. Data.** We conduct experiments where the comparison method (Veicht et al., 2024) is re-trained on the same dataset (Puffin-4M) as ours, strictly following its original training recipe. Interestingly, we find that the re-trained GeoCalib (Veicht et al., 2024) on our 4M dataset slightly underperforms the original GeoCalib model trained on its 40K dataset. The detailed evaluation results on the Puffin-*Und* test set are reported in Table A6.

By carefully analyzing these results, we offer two explanations for this phenomenon: (i) Model capacity *vs*. data scale. GeoCalib (Veicht et al., 2024) is a relatively lightweight CNN-like architecture with around 29M parameters. When trained on a significantly larger 4M-scale dataset with broad scene and distribution coverage, it tends to underfit: its limited capacity cannot fully model the entire distribution, so it only fits some sub-distributions well while inevitably neglecting others. (ii) Consistent observations from previous experiments. The GeoCalib authors report a similar trend in their ablation: when training the network on a 5× larger dataset, the camera understanding performance slightly degrades on most benchmarks, and no clear improvement is observed; they also show that more advanced architectures can further boost performance. These observations are fully consistent with our findings.

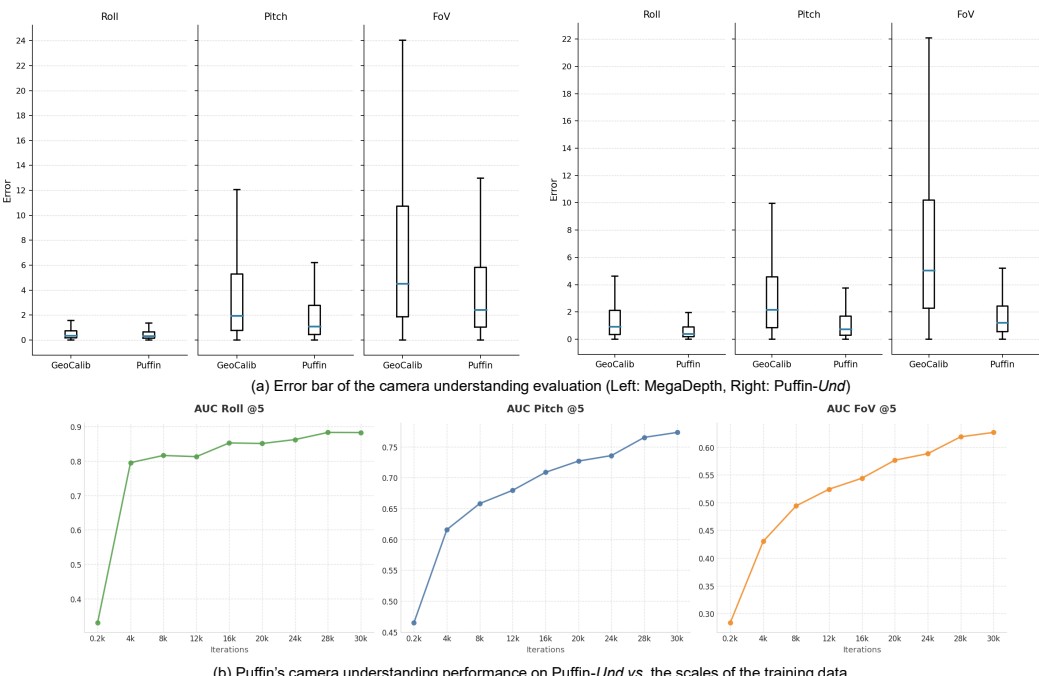

(a) Error bar of the camera understanding evaluation (Left: MegaDepth, Right: Puffin-*Und*)

(b) Puffin's camera understanding performance on Puffin-*Und vs*. the scales of the training data

**Figure A6: More data analysis of the experiments.** (a) We show the error bar of the camera understanding evaluation on GeoCalib (Veicht et al., 2024) and Puffin. (b) We show how the performance of Puffin scales with data size across different camera parameters.

In contrast, Puffin is built on a high-capacity LLM backbone, which, like other large multimodal models (*e.g.*, LLaVA, Qwen-VL, InternVL), requires sufficiently large and diverse training data

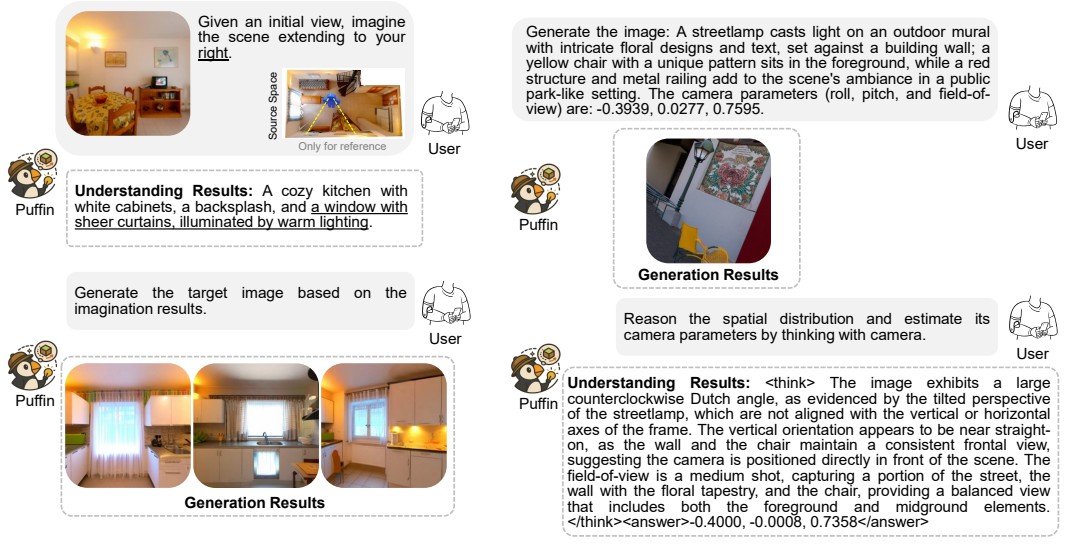

**Figure A7: Multi-turn interleaved capability of Puffin.**

to avoid overfitting and to learn an accurate joint distribution over images, camera geometry, and language. In this sense, Puffin-4M is not merely a "bonus", but rather a necessary data regime for such a unified multimodal model that jointly supports both understanding and generation.

Furthermore, we also conduct additional experiments by re-training our model on Puffin-4M of different scales (Stage-II SFT). As shown in Figure A6 (b), we observe clear and consistent improvements as the data scale increases. Overall, these results suggest that the dataset and the model contribute jointly and should not be viewed in isolation.

Beyond the above scaling results in Figure A6 (b), we find an interesting difference in trends across camera parameters. For roll, the model learns quickly even with relatively limited data, since it mainly relies on low- to mid-level geometric cues that are easy to capture (*e.g.*, strongly slanted lines indicating a large roll). By contrast, estimating pitch and FoV requires more holistic and high-level spatial understanding, which cannot be sufficiently captured by local visual patterns alone and therefore benefits more from larger-scale data to form robust spatial reasoning concepts. This observation is consistent with our discussion in Section A5.1. Based on these trends, we believe that further scaling the dataset would bring additional gains in camera geometry understanding, especially for pitch and FoV.

**Multi-turn Interleaved Capability.** Exploring the multi-round conversational capability of a unified multimodal model is meaningful. To this end, we conduct experiments on multi-turn interleaved conversations (generation → understanding and understanding → generation). As shown in Figure A7, Puffin can carry out coherent interleaved dialogues conditioned on previous turns. Specifically, it produces consistent cross-view generation results based on its previous reasoning, and accurate camera understanding based on its own generated images. This demonstrates that Puffin not only supports both capabilities within a single framework but can also use them in an interactive way over multiple conversational rounds, without any task-specific switching or separate models.

## A6   LIMITATION AND FUTURE WORK

Because our training dataset is constructed at a fixed resolution of $512 \times 512$, Puffin's image generation is currently restricted to a single scale. For camera understanding, we applied central cropping followed by resizing (Section 3.1), an operation that may discard semantically valid content and degrade performance, particularly when the aspect ratio deviates significantly from unity. While these limitations are orthogonal to our main focus, they could be addressed in future work by building multi-scale training datasets. Beyond data design, our evaluation of camera-controllable generation relied on an offline vision-based calibration method (Veicht et al., 2024). Although this approach reflects the best available practice, the calibration errors it reports can be ambiguous, especially for

generated images exhibiting only subtle spatial differences. Accurately evaluating spatial simulation thus remains an open challenge and is crucial for advancing camera-controllable generation. We plan to address this by incorporating stronger camera understanding models as evaluators and by designing benchmarks that more precisely capture geometric consistency. In addition, we aim to further enhance Puffin's cross-view capability and extend it to camera-centric video generation and understanding, paving the way for broader applications in dynamic and immersive environments.

Another interesting future direction is exploring camera-specific tokenizers, which could enable Puffin to model a richer and more nuanced camera control space beyond roll/pitch/FoV. Recent work such as Stable Cinemetrics (Chatterjee et al., 2025) provides a structured, industry-grounded taxonomy of cinematic controls, including a dedicated camera hierarchy with fine-grained nodes, and shows that even strong generative models still struggle to follow these nuanced camera instructions. This taxonomy and its evaluation protocol suggest a principled way to expand our photographic term set and to design more specialized camera tokens/bins that systematically cover lens types, shot grammar, and motion styles. Building on this insight, we plan to incorporate taxonomy-guided camera tokenization and supervision in the next iteration of Puffin, together with an extended Puffin-4M covering broader cinematic camera attributes, and release the resulting dataset and models to facilitate research on fine-grained multimodal spatial intelligence.

## A7   STATEMENTS

**Use of Large Language Models.** We follow ICLR's policy and use LLMs solely as general-purpose assistive tools. In our work, they were employed to polish the writing of certain paragraphs. They were not used for research ideation, experimental design, or the development of core technical contributions.

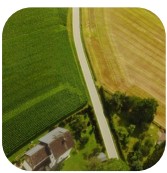 **<think>** The image exhibits a large counterclockwise Dutch angle, as evidenced by the diagonal orientation of the road and the fields, which are not aligned with the image's edges. The significant tilt-down perspective is apparent from the elevated view of the house and the surrounding landscape, providing a clear overhead look at the structures and fields. The close-up field-of-view is indicated by the detailed visibility of the house's roof and the individual trees, suggesting a focused and narrow scope that emphasizes specific elements within the scene. **</think><answer>**-0.4000, -0.6250, 0.4800**</answer>**

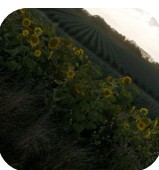 **<think>** The image exhibits a large counterclockwise Dutch angle, as evidenced by the tilted horizon line and the diagonal alignment of the rows in the field, which suggests a significant rotation of the camera. The vertical orientation shows a small tilt-down, indicated by the perspective that places the foreground sunflowers closer to the viewer while maintaining a clear view of the background, suggesting a slight downward angle without a dramatic shift. The medium shot field-of-view is apparent from the balanced composition that captures both the detailed foreground of sunflowers and the expansive background of the field. **</think><answer>**-0.6400, -0.2369, 0.6138**</answer>**

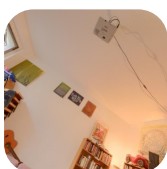 **<think>** The image exhibits a significant counterclockwise rotation, as evidenced by the tilted alignment of the bookshelf and the wall-mounted objects, which appear to lean towards the left side of the frame. The large tilt-up is apparent from the perspective, where the ceiling and upper wall are prominently visible, while the floor and lower elements are minimally shown, suggesting the camera is angled upwards. The wide-angle field-of-view is indicated by the broad capture of the room's interior, including multiple elements like the bookshelf, wall art, and guitar, all within a single frame, which emphasizes the expansive nature of the shot. **</think><answer>**-0.6000, 0.5814, 1.5348**</answer>**

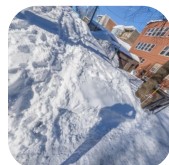 **<think>** The image exhibits a significant counterclockwise Dutch angle, as evidenced by the tilted alignment of the buildings and the snowbank, which appear to lean towards the left side of the frame. The large tilt-down orientation is apparent from the perspective, where the ground and snowbank dominate the foreground, while the buildings are positioned higher in the frame, suggesting a downward-angled view. The ultra-wide-angle field-of-view is indicated by the expansive capture of the snowbank and the surrounding environment, with noticeable distortion at the edges, particularly in the curvature of the snowbank and the buildings, which emphasizes the broad scope of the scene. **</think><answer>**-0.5364, -0.4957, 1.7608**</answer>**

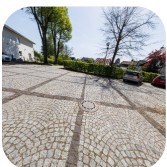 **<think>** The image exhibits a small counterclockwise Dutch angle, as evidenced by the slight diagonal alignment of the cobblestone pattern and the parked cars, which are not perfectly parallel to the edges of the frame. The vertical orientation shows a small tilt-down, indicated by the horizon line being slightly above the center of the image, suggesting a downward perspective. The ultra-wide-angle field-of-view is apparent from the expansive view of the cobblestone area, the trees, and the surrounding buildings, which are captured with significant distortion at the edges, particularly noticeable in the curvature of the cobblestone pattern and the elongated appearance of the trees and cars. **</think><answer>**-0.1460, -0.3487, 1.8088**</answer>**

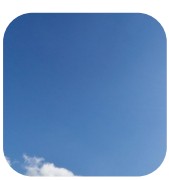 **<think>** The positioning of the clouds, which are not centered but rather skewed towards the bottom left, implies a large counterclockwise Dutch angle, as the frame is rotated to emphasize the diagonal orientation of the sky. The absence of any horizon line or terrestrial features indicates a significant tilt-up, as the camera is pointed upwards to capture the vastness of the sky. *The medium shot field-of-view is evident from the* balance between the open sky and the presence of the clouds, *providing a sense of depth without including too much of the surrounding environment.* **</think><answer>**-0.6704, 0.7098, 0.7048**</answer>**

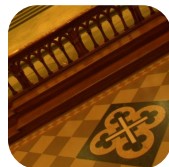 **<think>** The image exhibits a large clockwise Dutch angle, as evidenced by the diagonal orientation of the staircase and the railing, which are not aligned with the vertical or horizontal axes of the frame. The significant tilt-down perspective is apparent from the low angle at which the floor and the ornate emblem are captured, emphasizing the depth and leading the viewer's eye downward. The medium shot field-of-view is indicated by the inclusion of both the detailed emblem on the floor and the upper portion of the staircase, providing a balanced view that captures the architectural elements without excessive cropping or wide expanse. **</think><answer>**0.5280, -0.4158, 0.6660**</answer>**

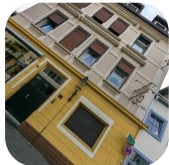 **<think>** The image exhibits a large counterclockwise Dutch angle, as evidenced by the diagonal alignment of the building's facade and the street, which tilts the scene to the left. The vertical orientation shows a significant tilt-up, as the lower portion of the building and street are cropped out, emphasizing the upper levels and the sky. The wide-angle field-of-view is apparent from the broad perspective that captures multiple stories of the building and the surrounding environment, including the adjacent structures and street elements, all within a single frame. **</think><answer>**-0.7119, 0.3745, 1.4028**</answer>**

**Figure A8: Visualization on our spatial reasoning process for camera understanding.** We highlight the reasoned spatially grounded visual cues regarding each camera parameter using different colors: roll, pitch, and FoV.

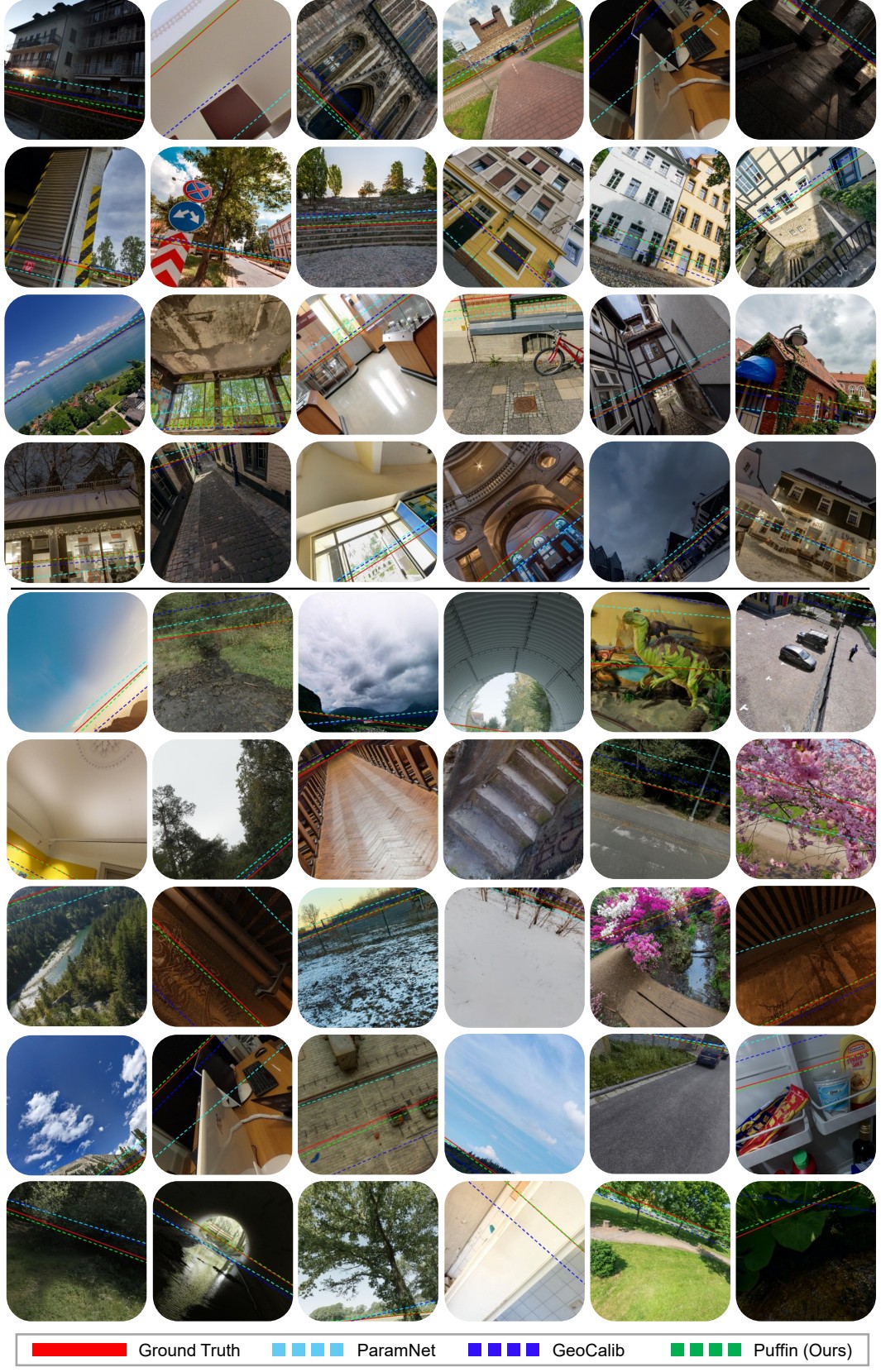

**Figure A9: Qualitative evaluations on the camera understanding methods with horizon line visualization.**
We show the common cases (with architectures or indoor) and challenging cases (with few geometric features or significant tilted camera poses) at the top and bottom, respectively.

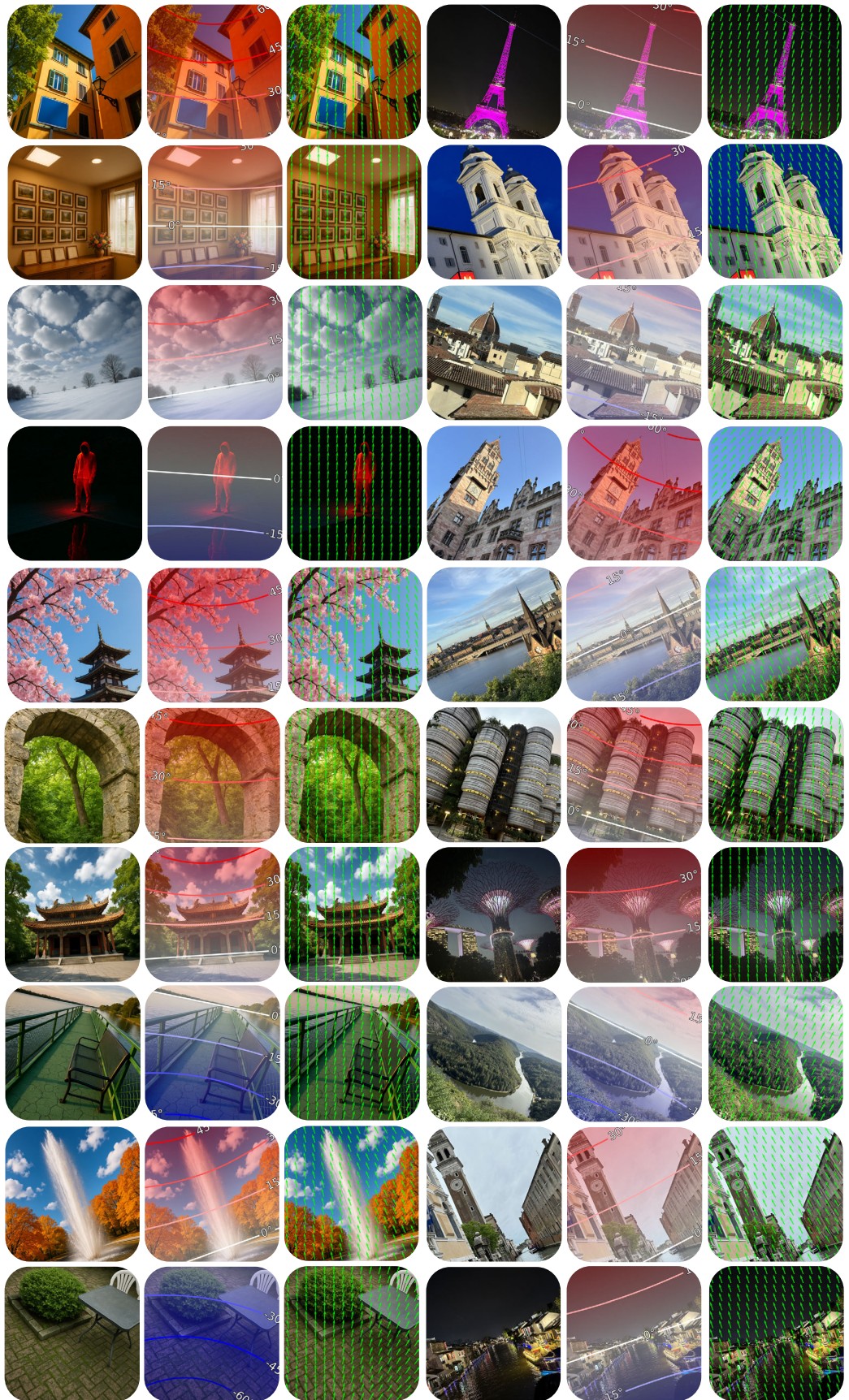

**Figure A10: Our camera understanding on AIGC images (OpenAI, 2025) (left) and real-world photographs (right).**

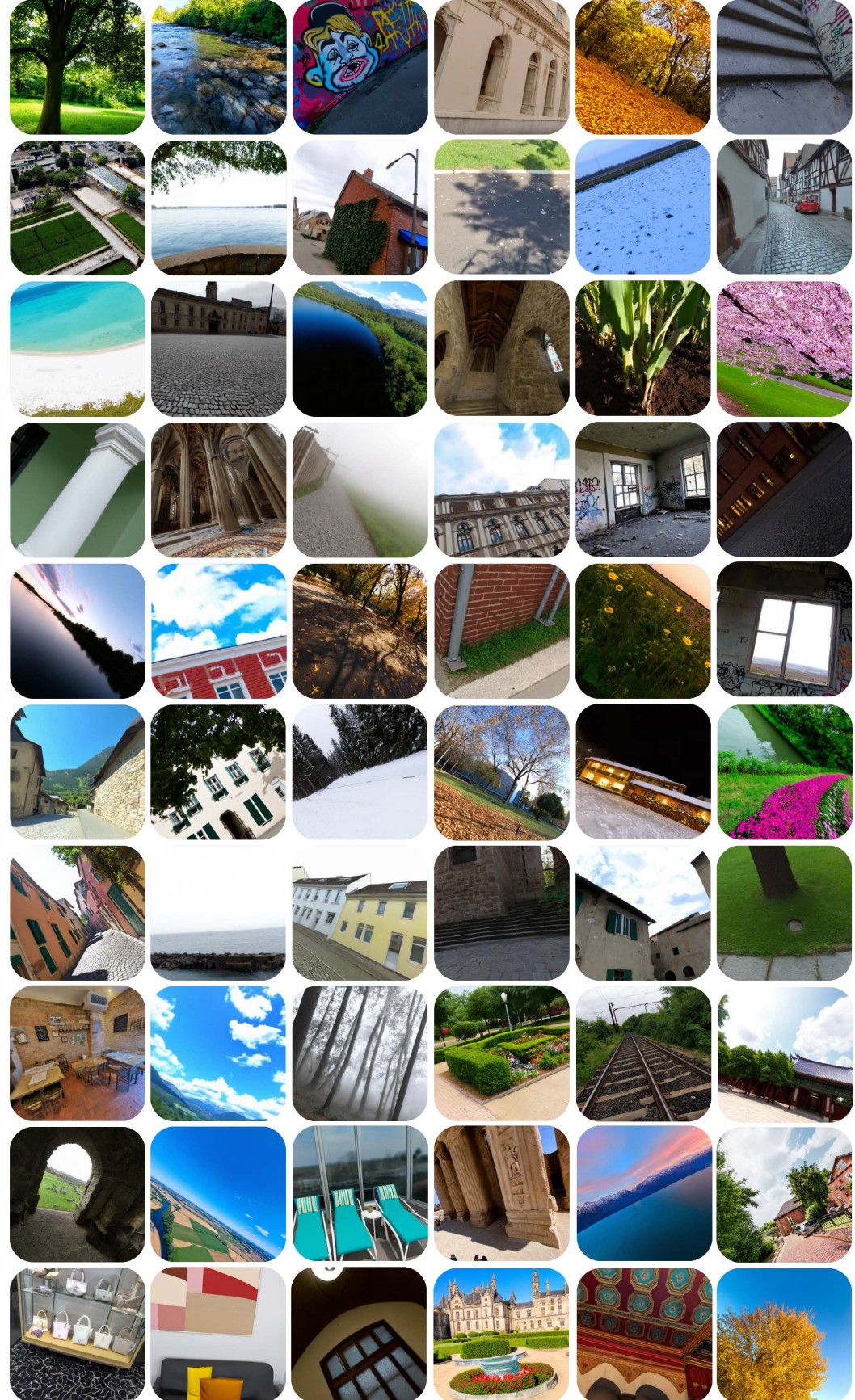

Figure A11: Our camera-controllable generation results with various camera configurations.

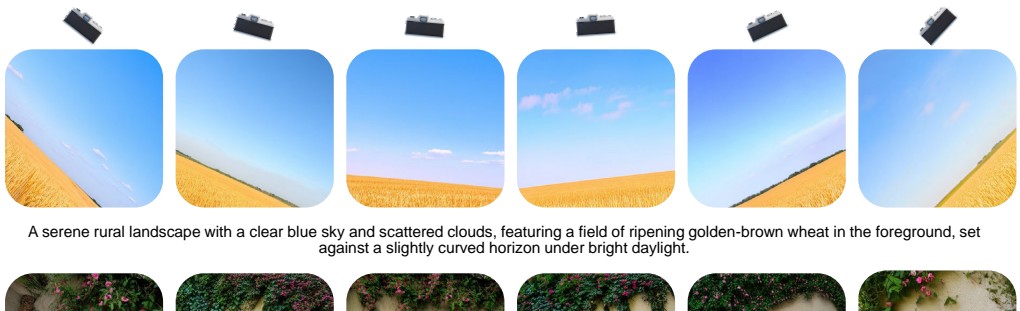

A serene rural landscape with a clear blue sky and scattered clouds, featuring a field of ripening golden-brown wheat in the foreground, set against a slightly curved horizon under bright daylight.

A brick pathway leads to a light-textured, vine-covered wall adorned with pink flowers, set against a backdrop of small plants, fallen autumn leaves, and a naturally overgrown, serene outdoor garden.

**(a)** Text-to-image generation with varying roll angles.

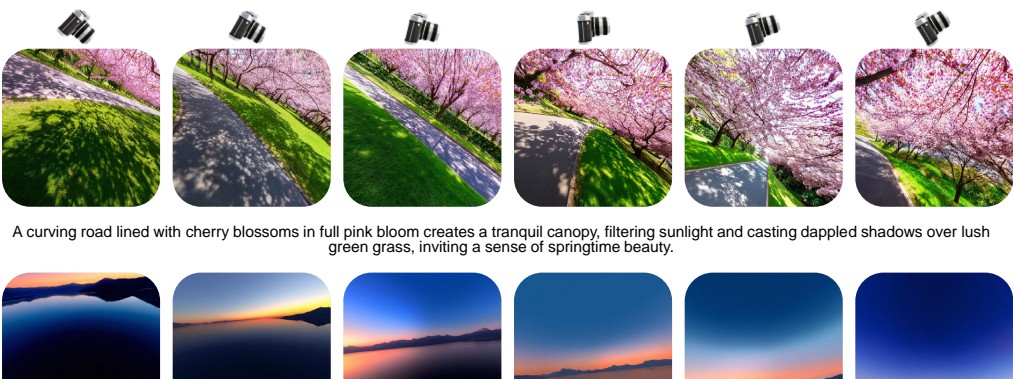

A curving road lined with cherry blossoms in full pink bloom creates a tranquil canopy, filtering sunlight and casting dappled shadows over lush green grass, inviting a sense of springtime beauty.

A serene sunset casts warm orange and pink hues over a vast lake, silhouetting distant mountains against a deep blue sky, exuding a tranquil and picturesque ambiance.

**(b)** Text-to-image generation with varying pitch angles.

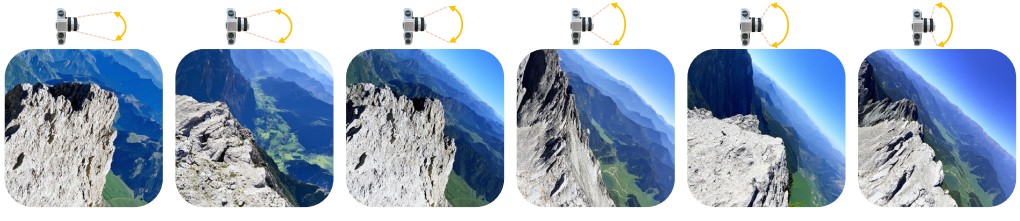

A rugged mountain peak overlooks a vast, green valley below, with sharp cliffs and distant, layered mountains stretching into a clear blue sky, emphasizing the isolation and dramatic beauty of the alpine landscape.

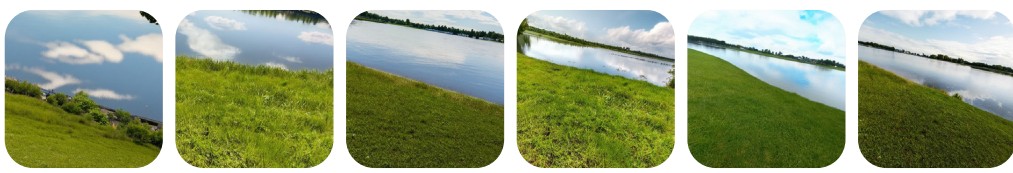

A serene lakeside scene with a grassy foreground sloping gently to a calm, cloud-dotted reflection in the water, framed by distant trees and structures, evoking a peaceful, rural atmosphere.

**(c)** Text-to-image generation with varying FoVs.

**Figure A12: Text-to-image generation (single-view) with specific controls for each camera parameter.**

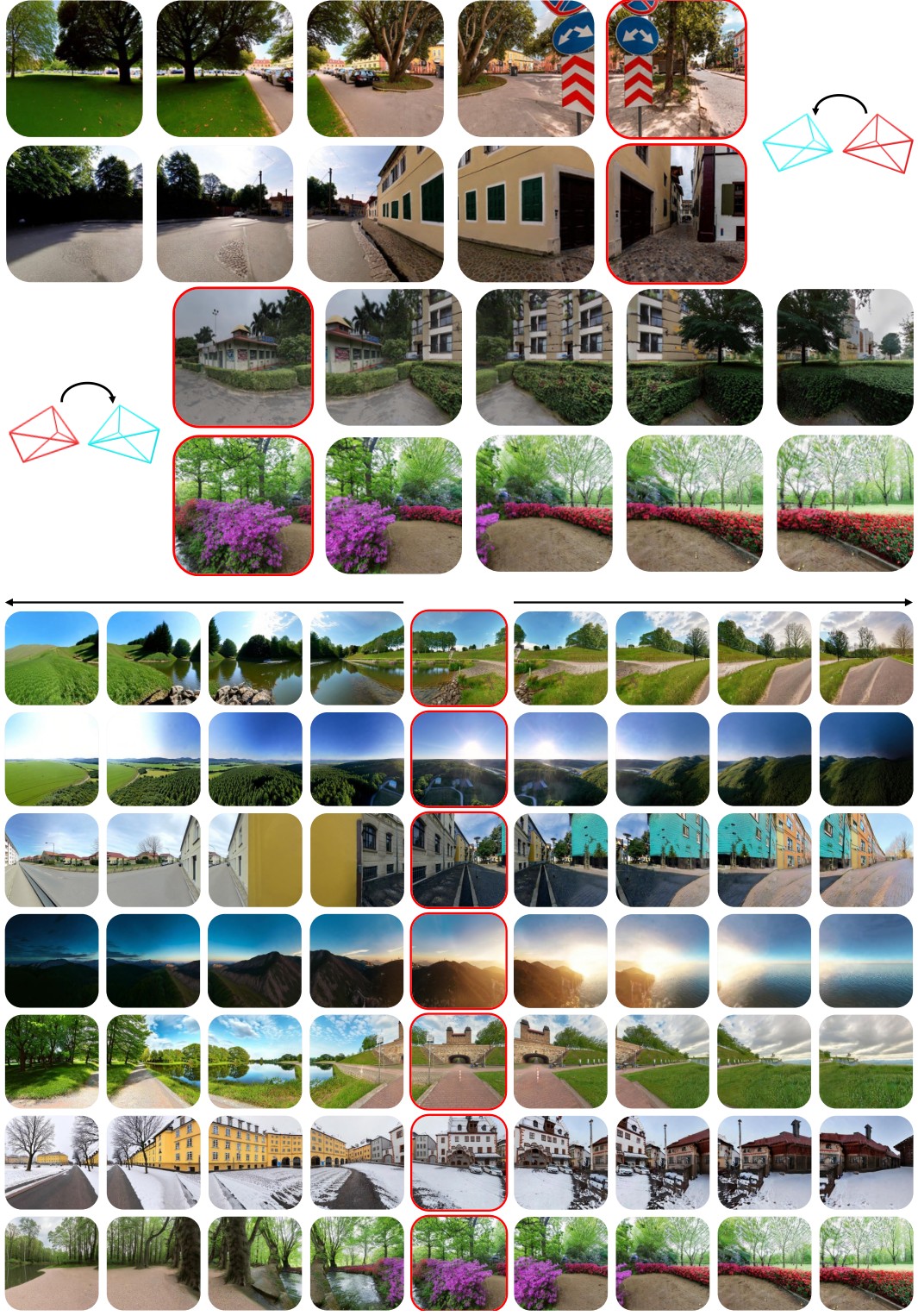

**Figure A13: Image-to-image generation (cross-view) with varying yaw angles.** The image with a red box denotes the initial view, and the others are the generated views based on the yaw deviation from the previous view.

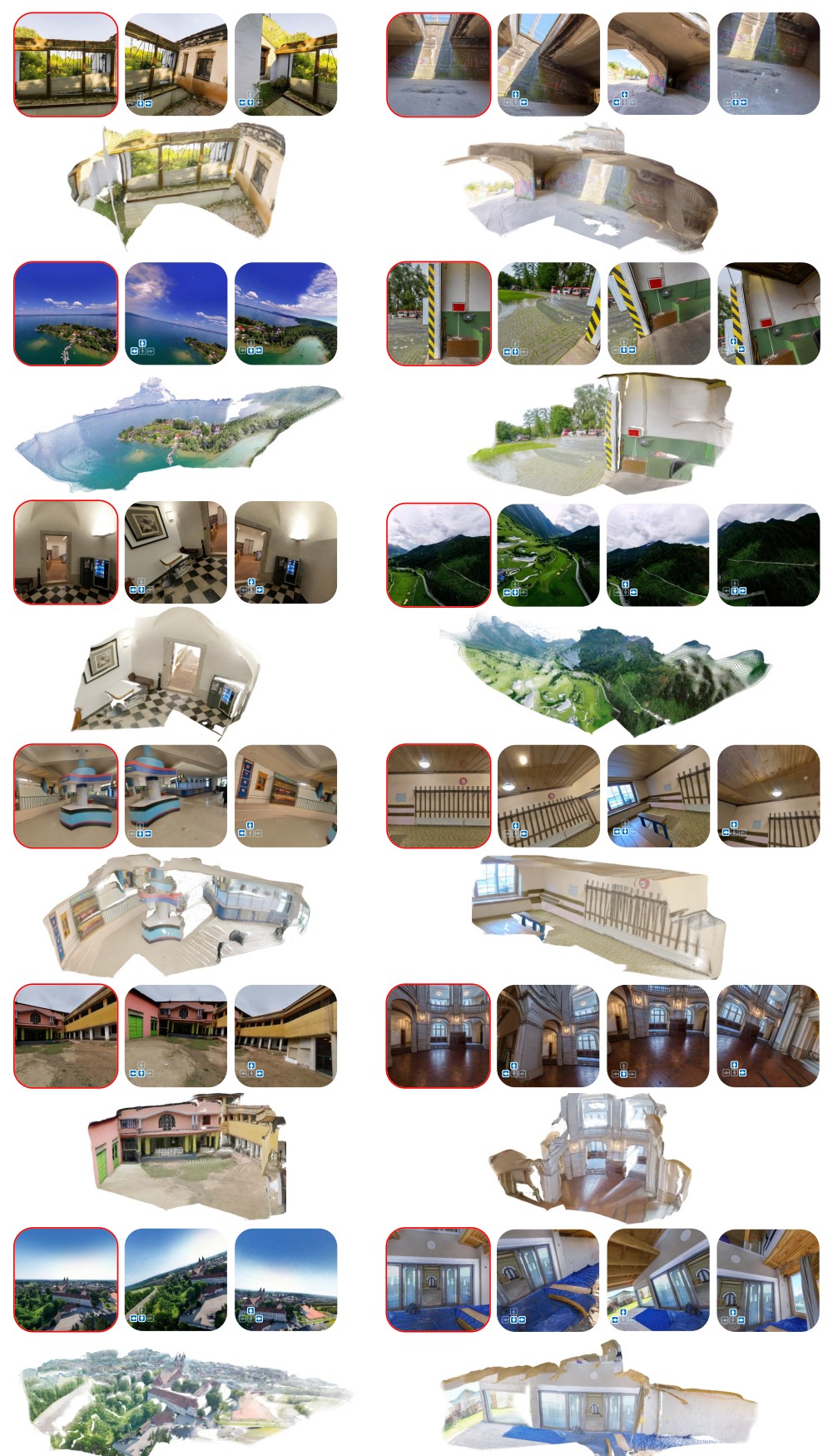

**Figure A14: World exploration results.** The 3D reconstruction results are obtained by VGGT.

**3D Space with Limited Focus**          **Spatial Imagination**

**Prompt:** Given an initial view in the environment, imagine the scene extending to your right.

**Imagination Description:** A cozy kitchen with white cabinets, a tiled backsplash, and a window with sheer curtains, illuminated by warm lighting.

Reference Image

**Prompt:** Given an initial view in the environment, imagine how the scene extends behind you.

**Imagination Description:** A living room with a fireplace, a flat-screen TV on a wooden stand, and a dining area visible through an open doorway.

Reference Image

**Prompt:** Given an initial view in the environment, imagine how the scene extends behind you.

**Imagination Description:** A living room with a white couch, a small table, and a large window offering a view of a residential area outside.

Reference Image

**Prompt:** Given an initial view in the environment, imagine the scene extending to your right.

**Imagination Description:** A group of people is gathered in a room with a large window, some standing and others seated on the floor, engaged in conversation or activities.

Reference Image

**(a)** Spatial imagination. The plausible imagination results are highlighted.

Input

Deviation (Yaw/Pitch)

| +0.04 | +0.0 | +0.08 | -0.04 | -0.12 |
| +0.26 | +0.0 | +0.17 | -0.21 | -0.05 |

Reference

Input

Deviation (Yaw/Pitch)

| +0.04 | +0.10 | +0.0 | +0.11 | -0.13 |
| -0.16 | +0.19 | +0.0 | +0.01 | +0.01 |

Reference

**(b)** Photographic guidance. The suggested deviations of the camera parameters (yaw/pitch) are highlighted.

**Figure A15: Examples of the spatial imagination and photographic guidance.**

