# OpenReview forum: "Thinking with Camera: A Unified Multimodal Model for Camera-Centric Understanding and Generation"
_ICLR.cc/2026/Conference — ICLR 2026 Poster_

### Official Review · Reviewer_D4GA · 2025-10-30

**Soundness:** 3
**Presentation:** 4
**Contribution:** 3
**Rating:** 6
**Confidence:** 3

**Summary:**

The paper proposes Puffin, a unified multimodal framework that treats the camera as a first-class modality to both (1) understand camera geometry from images and (2) generate images under precise camera control. Key ideas include: “thinking with camera” (reasoning in photographic terms such as Dutch angle / tilt-up) to bridge numeric parameters and language; a geometry-aligned vision encoder; and a continuous camera latent via pixel-wise camera maps to condition a diffusion generator. A new Puffin-4M dataset (4M image–text–camera triplets) and evaluation sets (Puffin-Und, Puffin-Gen) are introduced. Puffin reports strong results on camera understanding across several datasets and large margins over LMM baselines and PreciseCam for camera-controllable generation; ablations suggest the “thinking” and camera-map latent help.

**Strengths:**

1. Precise, flexible camera control via discrete tokens plus a dense per-pixel camera map.


2. Unified “think-with-camera” design that improves both understanding (pose/FoV) and controllable generation, and the concept itself is novel.


3. Scales with large curated data and cleanly extends to new tasks and parameters.

**Weaknesses:**

1. Missing details on the construction of Puffin-Und and Puffin-Gen.

2. Training at fixed 512 and use of central crop + resize for non-square inputs degrades understanding on datasets like LaMAR; this is acknowledged but might affect claims of generality.

**Questions:**

1. Line 207, table A1 in the appendix should not be directly referred to in the main content.

2. How sensitive are results to the exact photographic term thresholds (Table A1)? Any continuous-to-discrete ablation or learned bins?


3. How does Puffin handle fisheye or smartphone ultrawide distortion at test time? Could the camera-map latent be extended with distortion fields?


4. If you remove LLM-generated “thinking” (or replace with noisy/short versions), how quickly do understanding/generation scores degrade? Any attempts at self-consistency or rationale-free training?


5. The abbreviation “FoV” was never introduced.

---

> ### Author Response · Authors · 2025-11-21
> **Response to Reviewer D4GA (Part 1/2)**
>
> > ### **W1. Missing details on the construction of Puffin-Und and Puffin-Gen.**
>
> Thank you for pointing this out. We apologize for not making it sufficiently explicit in the original submission. Puffin-Und and Puffin-Gen are constructed by exactly the same pipeline as Puffin-4M; the only difference lies in the source images and the task-specific splits. In particular, the choices of camera-parameter ranges, the procedure for annotating the scene description, and the generation of pixel-wise camera maps all strictly follow the construction details described for Puffin-4M. Following your suggestion, we have highlighted this point more clearly in the revised version (Section A3) for Puffin-Und and Puffin-Gen.
>
> > ### **W2. Resolution issue in the dataset.**
>
> Thank you for highlighting this issue. We agree that training at a fixed 512 resolution and using central crop + resize can hurt performance on datasets like LaMAR, where resolution, aspect ratio, and boundary content carry important geometric cues. As also reflected in our results, Puffin performs strongly on other benchmarks (*e.g.*, MegaDepth, TartanAir, Puffin-Und), and the degradation is mainly concentrated on LaMAR’s more challenging imaging setup, so the issue is more about input format than the core camera-centric design. Furthermore, we are actively extending Puffin-4M to cover a broader range of resolutions, aspect ratios, scene distributions, image qualities, and task dimensions. In future work, we plan to release the expanded dataset and updated models to better support high-resolution and non-square camera-centric scenarios, and we hope this will further benefit the community.

---

> ### Author Response · Authors · 2025-11-21
> **Response to Reviewer D4GA (Part 2/2)**
>
> > ### **Q1. Table A1 in the appendix should not be directly referred to in the main content.**
>
> We appreciate the reviewer’s valuable suggestion. We have revised the main text so that it no longer directly references “Table A1”; instead, we briefly summarize the key content in the main text and refer more generally to the Appendix for full details.
>
> > ### **Q2. Photographic term thresholds (Table A1) and any continuous-to-discrete ablation or learned bins?**
>
> Thank you for this insightful question. Table A1 is mainly intended to bridge textual photographic terms and numeric camera parameters. The thresholds there are currently set heuristically based on the physical meaning of each term, which gives a simple and interpretable discretization.
>
> Importantly, our camera understanding is not trained as a pure discrete classification over these bins. The LLM part predicts next tokens with a cross-entropy loss over (i) the coarse photographic terms (Dutch angle, tilt-up/down, etc.) and (ii) the fine-grained numeric camera parameters. In other words, the reasoning first produces a linguistically coarse description, and then, conditioned on this reasoning, predicts continuous-valued roll/pitch/FoV tokens. These two levels are learned jointly rather than in isolation, and the final supervision on the numeric parameters actually acts as a set of fine-grained bins, which reduces sensitivity to the exact coarse thresholds in Table A1.
>
> > ### **Q3. How does Puffin handle fisheye or smartphone ultrawide distortion at test time? Could the camera-map latent be extended with distortion fields?**
>
> At the current stage, we candidly acknowledge Puffin does not explicitly handle fisheye or smartphone ultrawide distortion at test time, mainly due to a training dataset design limitation: the perspective image in Puffin-4M is rendered under a standard pinhole camera model without radial distortion. We have also acknowledged this in Section A.3.2, where we noted that incorporating radial distortion via an additional parameter *k* is left as future work.
>
> For the framework itself, however, Puffin is quite amenable to distortion modeling without changing the core architecture:
>
> - Camera tokens can be easily extended by appending one (or several) distortion parameters after the current roll/pitch/FoV tuple.
>
> - Camera-map latents can likewise be flexibly extended with distortion fields: given a fisheye camera model (*e.g.*, radial distortion with parameter *k*), we can offline project each pixel through the formulation of Perspective Field [1] to produce distortion-aware latitude/up-vector maps. These can be fed into the diffusion backbone as additional channels under the same configuration with the current standard camera model.
>
> Following your suggestion, we plan to extend Puffin-4M to include fisheye and smartphone ultrawide imagery with explicit distortion parameters, and to release the corresponding dataset and updated models in future work.
>
> [1] Jin, Linyi, et al. "Perspective fields for single image camera calibration." Proceedings of the IEEE/CVF Conference on Computer Vision and Pattern Recognition. 2023.
>
> > ### **Q4. The effectiveness of the thinking with camera.**
>
> Thank you for this thoughtful question. We would like to clarify that we have included an ablation along this line in the paper: Section 3.3 and Table 3 compare Ours w/Thinking (with LLM-generated reasoning) against Ours (without “thinking” supervision, *i.e.*, a rationale-free setting). The results show that the Thinking design consistently improves performance, especially for pitch and FoV prediction, which rely more on broader contextual priors rather than purely local geometric cues. This suggests that learning from explicit “thinking with camera” rationales helps the model capture hierarchical spatial context beyond localized features.
> In our current submission, we have not yet included variants with deliberately noisy/short rationales or self-consistency–style training, but kindly note that Ours already corresponds to a rationale-free baseline. As suggested, we will more clearly highlight in the revision that this ablation directly measures the impact of LLM-generated “thinking”, and we view exploring noisy/perturbed rationales and self-consistency mechanisms as a valuable direction for future work.
>
> > ### **Q5. The abbreviation FoV was never introduced.**
>
> Thank you for your thorough review. We have updated the full term “Field-of-View (FoV)” at its first occurrence in the revised paper.

---

> > ### Comment · Reviewer_D4GA · 2025-11-25
> >
> > Thanks for your detailed rebuttal which cleared out my questions. As reviewer 2Lvj suggests, an interesting extension of this work is to apply RL training.
> >
> > There's a writing issue which should be fixed. In the current manuscript, some figures in appendix are directly referred without mentioning the section number (e.g. line 217, line 373). You can correct this by mentioning appendix section number, and mentioning the specific figure in that appendix section. Doing so ensures that the paper complies with the 9-page rule.

---

> > > ### Author Response · Authors · 2025-11-25
> > > **Response to Reviewer D4GA**
> > >
> > > Thank you very much for your follow-up and for carefully pointing out this writing issue. We have revised all instances where appendix figures were directly referenced (including around Lines 217 and 373) so that they now cite the corresponding appendix section number and figure number. These changes are highlighted in red in the revised manuscript and help ensure that the paper complies with a reasonable structure.
> > >
> > > We hope that these revisions fully address your concerns, and we would be grateful if you could kindly raise your score. Thank you again for your time and thoughtful review of our work.

---

### Official Review · Reviewer_2Lvj · 2025-10-31

**Soundness:** 3
**Presentation:** 3
**Contribution:** 3
**Rating:** 8
**Confidence:** 4

**Summary:**

The paper focuses on an interesting task -- unified camera-centric understanding and generation. There are two major contributions: they collect a 4M high-quality dataset with multiple labels for both mm understanding and generation, while they also train a unified VLM called Puffin for the target task. The proposed model achieved state-of-the-art performance on the camera-centric tasks and enables quite a few interesting applications. Overall, this paper positions “thinking with camera” as a step toward foundation‑level models that natively incorporate 3D geometry in both understanding and generation.

**Strengths:**

- This paper is well-structured and easy to follow
- The framing of a single model and interface for both camera‑centric understanding and generation, with camera tokens as the key abstraction, is interesting (though it follows the popular way of designing unified models)
- Puffin‑4M appears carefully constructed to supervise geometry/camera attributes across many scenes.
- Superior performance compared with sotam odels, for both tasks.

**Weaknesses:**

- Missing specialized strong baselines: some comparisons to **strongest specialized 3D models** (e.g., recent camera calibration/pose methods) seem limited in the main text; more head‑to‑head numbers would strengthen claims.

- L408-416: for generation comparisons, it is also useful to show standard generation evaluation metrics, such as FID -- perfect camera control makes no sense if the overall visual quality is poor.
- Better to show the error bar for the compared models in the major experiments.
- Details of thinking with camera is missing.

**Questions:**

- In A.3.3, the caption of the Puffin is generated by Qwen2.5VL will this explain why Qwen-image (reuse qwen2.5vl as encoder) performs better than GPT4o and Nano-banana? How to mitigate the affect of this issue?
- Table A.3, bottom right cell, should be 0.2 rather than 0.05?
- For thinking with camera, will RL, such as GRPO, help improve the reasoning capability?
- Following the previous comment, it seems the thinking with camera training data is generated by Qwen2.5VL, how to make sure Qwen is able to provide some high-quality data since it is not trained on this task? Is it possible to measure their quality?
- Are **camera tokens** robust to out‑of‑distribution intrinsics/extrinsics?
- How does performance scale with data set size and quality?

---

> ### Author Response · Authors · 2025-11-21
> **Response to Reviewer 2Lvj (Part 1/3)**
>
> > ### **W1. Missing specialized strong baselines.**
>
> Thank you for pointing this out. For the camera understanding setting, we compared against, to the best of our knowledge, the strongest single-image camera calibration methods, including DeepCalib, CTRL-C, MSCC, ParamNet, SVA, UVP, and in particular GeoCalib. GeoCalib is the current state-of-the-art approach for joint intrinsic–extrinsic estimation on the standard benchmarks, and we strictly followed its evaluation protocol (datasets, splits, and metrics) for all our head-to-head results. For camera-controllable generation, we also included PreciseCam as the strongest task-specific baseline.
> If you are aware of any more recent or stronger single-image calibration methods that fit our experimental setting and are publicly accessible, we would be very grateful if you could kindly let us know. We would be more than happy to include them in the final version of the paper.
>
> > ### **W2. Generation evaluation metrics.**
>
> We agree that reporting standard generation metrics is meaningful. Following your thoughtful suggestion, we have evaluated all comparison methods using FID; please refer to the table below for details.
>
> | | GPT-4o* | GPT-4o+ | Qwen-Image* | Qwen-Image+ | Nano-Banana* | Nano-Banana+ | PreciseCam | Puffin |
> | :--- | :---: | :---: | :---: | :---: | :---: | :---: | :---: | :---: |
> | **FID** | 95.92 | 94.43 | 83.31 | 83.37 | 91.66 | 88.02 | 90.91 | 69.46 |
>
> The results show that our method not only achieves the best performance on camera-controllable generation (*i.e.*, the original spatial simulation evaluations), but also delivers competitive visual quality, attaining the lowest FID among all models. The relatively high FID values across all methods are mainly due to the limited size and distribution of the test set (650 images). Nevertheless, our approach consistently improves over all baselines. We have incorporated this visual quality evaluation into Table 2 in the revised paper.
>
> > ### **W3. Error bar for the compared models.**
>
> We appreciate the reviewer’s valuable suggestion. We would first like to clarify that our primary metric is the AUC over recall thresholds (1°, 5°, 10°). This serves a role similar to an error bar by summarizing robustness across different error tolerances, characterizing performance over the entire recall–threshold curve rather than at a single operating point.
> Following your suggestion, we have also added error bars for the representative baselines and our method in Figure A6(a) of the revised paper to provide a clearer and more comprehensive comparison. The results show that our method exhibits better robustness across the entire data distribution. The improvement is consistent across all intrinsics/extrinsics components and remains stable even under challenging camera configurations.
>
> > ### **W4. Details of thinking with camera is missing.**
>
> Thank you for raising this point. In Section 2.1 (Lines 187–217), we provide a detailed explanation of thinking with camera, including its three core components: Spatially Grounded Visual Cues, Professional Photographic Terms, and Geometric Context. Additionally, we also illustrate the training details of thing with camera in Section A4 (Lines 1263-1269).
>
> If there are specific aspects of thinking with camera that you would like to see elaborated, we would be glad to further expand or clarify them in the final version.

---

> ### Author Response · Authors · 2025-11-21
> **Response to Reviewer 2Lvj (Part 2/3)**
>
> > ### **Q1. Qwen-image performance.**
>
> This is a very interesting question. We would like to address it from two perspectives:
>
> - The caption for camera-controllable generation is composed of two parts: a scene caption and a camera caption. The scene caption is generated by Qwen2.5-VL, while the camera caption is randomly sampled from a standard camera model. Since our evaluation of generation mainly relies on geometry-related metrics derived from the camera parameters (*e.g.*, Up vector, Latitude, Gravity), the results are largely insensitive to the scene captions that describe image content.
>
> - Qwen-image does not consistently outperform other large models in spatial simulation. For example, for the mean latitude error (lower is better), Qwen-image obtains 15.92 compared to 14.67 for GPT-4o and 15.80 for Nano-banana. This behavior suggests that our protocol does not systematically favor Qwen-image, and that the evaluation fairly reflects spatial simulation performance across different methods.
>
> > ### **Q2. Details of the data sampling ratio in Table A.3.**
>
> We sincerely thank you for checking this detail. The bottom-right cell in Table A.3 is not a typo: the original sampling ratio is 8:8:1 across three types of data, which we approximated in the paper as 0.4:0.4:0.05. To avoid confusion and potential misinterpretation (*e.g.*, expecting 0.2), we have updated the table in the revised version to report the ratio 0.47:0.47:0.06 directly.
>
> > ### **Q3. Will RL further enhance thinking with camera?**
>
> This is a great direction and we agree that RL (*e.g.*, GRPO-style training) is likely to further enhance thinking with camera. At present, Puffin’s reasoning is learned purely from supervised signals (camera annotations + language supervision). In principle, we can augment this with camera-geometry–aware rewards during training, such as penalties based on roll/pitch/FoV errors, consistency of multi-view reasoning, or correctness of qualitative photographic terms. This is in line with recent advances in spatially oriented LMMs that use RL to sharpen reasoning. We consider integrating RL with geometry-specific rewards into Puffin as an exciting avenue for future work.
>
> > ### **Q4. Quality measurement of the thinking caption.**
>
> We are grateful for your thoughtful question. In the following, we address it from three perspectives:
>
> - For captioning the thinking with camera, we deliberately use Qwen2.5-VL-32B to generate reasoning annotations, while the ordinary scene captions are produced by a lighter 7B model. This design trades off labeling cost and quality: the more challenging spatial–geometric reasoning is handled by a stronger backbone, which empirically yields more reliable descriptions.
>
> - We conduct a user study by randomly sampling 200 reasoning annotations and asking human evaluators to judge whether the textual reasoning is consistent with the image’s spatial geometry (*e.g.*, horizon, tilt, FoV). Around 95% of the sampled annotations are rated as geometrically consistent, with the few inaccurate cases mainly occurring in challenging scenarios such as textureless sky. This suggests that Qwen2.5-VL-32B provides generally reliable spatial reasoning when guided by our photographic prompting strategy.
>
> - The effectiveness of these reasoning labels is further supported by our ablation in Table 3, where Ours w/Think consistently outperforms Ours (without thinking) on all three camera parameters (roll, pitch, FoV). This performance gap suggests that the thinking with camera annotations is not only of sufficient quality, but also provides meaningful additional supervision that improves camera understanding.

---

> ### Author Response · Authors · 2025-11-21
> **Response to Reviewer 2Lvj (Part 3/3)**
>
> > ### **Q5. Are camera tokens robust to out‑of‑distribution intrinsics/extrinsics?**
>
> Thank you for the question. To address it clearly, we would like to distinguish between the representation capacity of camera tokens and the generalization of the trained model:
>
> - Our camera tokens are constructed from continuous roll, pitch, and FoV values within *bounded ranges* (roll ∈ [−90°, 90°], pitch ∈ [−90°, 90°], FoV ∈ [0°, 180°]). The tokenizer can represent any value in these intervals, so there is no inherent out-of-distribution issue at the token level as long as the parameters lie within these bounds.
>
> - Our parameter ranges of the training dataset strictly follow the state-of-the-art GeoCalib and other representative single-image calibration work, which report that these intervals cover the vast majority of common scenarios. Our training data is constructed within the same ranges to ensure fair comparison and realistic coverage.
>
> - In our tests with more extreme camera settings beyond the dominant training distribution, Puffin remains reasonably robust for moderate deviations but degrades when parameters deviate too far from the dataset ranges. We view this as a limitation of data coverage and training, rather than of the camera-token design itself. In principle, extending Puffin-4M to include broader intrinsics/extrinsics distributions would directly improve robustness to such OOD settings without changing the tokenization scheme.
>
> > ### **Q6. How does performance scale with data set size and quality?**
>
> Following your insightful suggestion, we conducted additional experiments by re-training our model on Puffin-4M of different scales (Stage-II SFT). As shown in Figure A6(b) of the revised paper, we observe clear and consistent improvements as the data scale increases. Puffin is built on a high-capacity LLM backbone, which, like other large multimodal models (*e.g.*, LLaVA, Qwen-VL, InternVL), requires sufficiently large and diverse training data to avoid overfitting and to learn an accurate joint distribution over images, camera geometry, and language. In this sense, Puffin-4M is not merely a “bonus”, but rather a *necessary data regime* for such a unified multimodal model that supports both understanding and generation.
>
> Moreover, we find an interesting difference in trends across camera parameters. For roll, the model learns quickly even with relatively limited data, since it mainly relies on low- to mid-level geometric cues that are easy to capture (*e.g.*, strongly slanted lines indicating a large roll). By contrast, estimating pitch and FoV requires more holistic and high-level spatial understanding, which cannot be sufficiently captured by local visual patterns alone and therefore benefits more from larger-scale data to form robust spatial reasoning concepts. This observation is consistent with our discussion in Section A5.1. Based on these trends, we believe that further scaling the dataset would bring additional gains in camera geometry understanding, especially for pitch and FoV.

---

> ### Comment · Reviewer_2Lvj · 2025-11-26
> **Post-rebuttal response**
>
> I thank the authors for the detailed responses. Most of my concerns have been resolved. I am learning to maintaining my original rating.

---

> > ### Author Response · Authors · 2025-11-27
> > **Resonse to Reviewer 2Lvj**
> >
> > We are glad to hear that our rebuttal has addressed most of your concerns, and we sincerely appreciate your decision to maintain your positive rating. The additional experiments and corresponding analyses you requested, as well as the related revisions, have all been incorporated into the updated version of the paper.
> >
> > Thank you again for your time and effort in reviewing our work.

---

### Official Review · Reviewer_j8cQ · 2025-11-01

**Soundness:** 3
**Presentation:** 4
**Contribution:** 4
**Rating:** 8
**Confidence:** 4

**Summary:**

This work presents Puffin, a multimodal model that unified camera-centric generation and understanding. Puffin treats camera parameters such as pitch, yaw and FoC as discrete tokens, similar to language tokens, thereby enabling thinking with cameras. Puffin-4M is introduced which consists of vision-language-camera triplets, constructed by collecting panoramic images followed by perspective crops on different camera angles, and synthetic captions generated by a VLM. The camera understanding takes as input : text and camera discrete tokens, and image tokens from a geometry-aligned vision encoder, while the camera generation module additionally has a learnable connector module and camera maps as additional conditioning. After multi-stage training on Puffin-4M, experiments demonstrate Puffin’s superior performance over specialized models for camera-centric generation and understanding. Futher, with instruction tuning Puffin can be extended to cross-view tasks such as world exploration and spatial imagination.

**Strengths:**

1. The paper is easy to read and well-motivated; in that it is the first attempt to unify camera generation and understanding.
2. Puffin outperforms existing baselines and methods across multiple benchmarks.
3. The finding that representing and learning camera parameters as discrete tokens is an impactul finding; making them almost analogous with how text tokens are used in today's generative vision systems.
4. The paper is technically dense; all design choices and the reasonings behind them are well documented.
5. I believe the Puffin-4M dataset will be a great contribution to the community.

**Weaknesses:**

I have a few minor weakeness/comments :

1. As shown in Table A1, the parameter-to-term mapping is not exhaustive; for example how is a small tilt-up with a clockwise
dutch angle handled?
2. Since, the camera parameter tokenizer is similar to the text tokenizer, is there any ablation that show the effect of different kinds of text encoder? For example, is there a performance delta in using encoder / encoder-decoder / decoder-only models?

**Questions:**

Please refer to weaknesses.

---

> ### Author Response · Authors · 2025-11-21
> **Response to Reviewer j8cQ**
>
> > ### **W1. The design of the parameter-to-term mapping.**
>
> Thank you for the insightful comment. In Table A1, we disentangle the parameter-to-term mapping per camera parameter (roll, pitch, FoV) rather than enumerating all their combinations. To jointly reason about multiple camera attributes, we can simply compose the corresponding photographic terms in a cascade-like language form. For example, a coherent spatial reasoning chain can be: *The image exhibits a large counterclockwise Dutch angle, as evidenced by ……. The significant tilt-down perspective is apparent from ……. The close-up field-of-view is indicated by ……*.
>
> More qualitative examples of such multi-attribute reasoning are provided in Figure A8 of the Appendix.
> For your mentioned case, a “small tilt-up with a clockwise Dutch angle” would be represented as a combination of two terms (*e.g.*, “slight tilt-up” + “slight clockwise Dutch angle”), rather than a single entry in Table A1, which is why the table focuses on per-parameter bins instead of enumerating all joint cases. Following your valuable suggestion, we have also clarified in the caption of Table A1 that combining different terms linguistically allows us to jointly describe the full camera geometry of an image.
>
>
> > ### **W2. The architecture of the camera parameter tokenizer.**
>
> We appreciate your thoughtful comments. In Section A2.1, we clarify that the camera parameter tokenizer follows exactly the same procedure as the text tokenizer in Qwen2.5. Since camera parameters are numerical, we first serialize them into discrete tokens, which are naturally handled by the standard text tokenizer without requiring any extra module. Introducing an additional tokenizer (or a separate text encoder) would substantially increase the alignment burden across modules and modalities in a unified multimodal model. For this reason, we keep the vanilla decoder-only LLM backbone and its tokenizer to process both language and camera parameters. Following your suggestion, we have added the above explanation in the revised paper. Exploring different kinds of text encoders is indeed an interesting direction, and we look forward to investigating it in future work.

---

> > ### Comment · Reviewer_j8cQ · 2025-11-21
> > **Thanks for your rebuttal**
> >
> > Thanks for the rebuttal. I appreciate the clarity around combinations of camera terms. I keep my rating.
> > I also believe that exploring camera specific tokenizers would be an interesting future direction as it would enable exploration into multiple nuances of camera, for example as enumerated here : https://stable-cinemetrics.github.io/.

---

> > > ### Author Response · Authors · 2025-11-22
> > > **Thanks for your response**
> > >
> > > Thank you for the update and for maintaining your positive rating of 8. We also really appreciate your recommendation of Stable Cinemetrics and the broader idea of camera-specific tokenization. Stable Cinemetrics introduces a structured, industry-grounded taxonomy of cinematic controls (including a dedicated Camera hierarchy) and a benchmark/evaluator that reveal substantial gaps in current generative models on fine-grained camera control.
> > >
> > > We see this as highly aligned with Puffin’s goal. In particular, their disentangled and hierarchical camera taxonomy suggests a principled way to (i) enrich our photographic term space beyond roll/pitch/FoV, and (ii) design camera-specific tokenizers/bins that cover nuanced controls (*e.g.*, lens types, motion styles, shot grammar) in a more systematic and scalable manner. We believe integrating such a taxonomy could help Puffin learn finer camera geometry/semantics and support broader cinematic reasoning and controllable generation.
> > >
> > > Building on your helpful suggestion, we have added Stable Cinemetrics to our discussion of the future work (see Line 1518-1528) and outlined plans to explore camera-specific tokenization and taxonomy-guided supervision/evaluation in the next iteration of Puffin.

---

### Official Review · Reviewer_pAbj · 2025-11-02

**Soundness:** 3
**Presentation:** 4
**Contribution:** 3
**Rating:** 6
**Confidence:** 4

**Summary:**

This paper proposes Puffin, a unified multimodal model for camera-centric understanding and generation. Specifically, Puffin formulates camera understanding as an AR text token generation task, and camera-centric generation as a learnable query + conditional diffusion generation task in a meta-query way. Both tasks are unified in an LLM with multiple additional components (a connector and diffusion model for generation, a visual encoder for understanding).  The authors further build a large-scale dataset, Puffin-4M, containing 4M vision-language-camera triplets, to facilitate this new paradigm. Extensive experiments on benchmarks demonstrate Puffin’s strong performance across both understanding and generation tasks.

**Strengths:**

1. Unified camera-centric understanding and generation framework: It's a novel and meaningful idea to unify camera understanding and controllable generation in one framework.
2. Thinking with Camera: Interpreting camera parameters as text description bridges geometry and language, and further enables reasoning with the camera in understanding and generation tasks.
3. The newly collected 4M high-quality camera-centric vision-language dataset should be very helpful to the community.
4. The proposed model can outperform previous works in both understanding and generation tasks, covering multiple benchmarks. Also, the ablation comprehensively covers different components of the model.

**Weaknesses:**

1. Parameter comparison over previous works:  A clear analysis of model scale and computational cost is missing. Since Puffin integrates multiple large-scale model components (LLM, diffusion model, vision encoder), comparing the total parameter count and FLOPs with prior understanding and generation baselines (e.g., GeoCalib, PreciseCam, etc) is needed.
2. Data vs. model contribution: It's not quite clear whether Puffin’s performance gains mainly come from the model architecture or the large new dataset (Puffin-4M). Ablation and fair comparison under the same training dataset would be useful.
3. Multi-round conversation capability: While the paper discusses instruction tuning and cross-view reasoning, it is not clear whether the proposed model can generalize well on multi-turn interleaved understanding and generation. For example, first do generation, then understanding in the second round.

**Questions:**

Please see weaknesses.

Besides, I have one more question regarding whether understanding and generation can benefit each other. In this paper, the author(s) claim that unifying understanding and generation can help each other (to a significant degree). However, it seems that in other general unified image understanding and generation works, the mutual benefit is not quite clear or not very significant. Can you explain why?

---

> ### Author Response · Authors · 2025-11-21
> **Response to Reviewer pAbj (Part 1/2)**
>
> > ### **W1. Parameter comparison over previous works.**
>
> We thank the reviewer for this suggestion. We agree that an explicit comparison of model size and computational cost will make the paper clearer, and we have added a new table in the Appendix. For your convenience, we also reproduce this table below.
>
> | | **GeoCalib** | **PreciseCam** | **Puffin** |
> | :--- | :--- | :--- | :--- |
> | **Type** | Specialized Model | Specialized Model | Unified Model|
> | **Parameters** | 28.9M | 1.3B | 4.4B |
> | **GFLOPs** | 3.47 × 10² | 2.67 × 10³ | 2.92 × 10⁵ |
>
> While Puffin is larger than previous specialized models, it replaces separate understanding and generation networks with a single unified model that handles both tasks within one framework. This design not only simplifies deployment, but also allows us to fully exploit a high-capacity backbone when training on large-scale multimodal datasets. In terms of overall parameter count and FLOPs, Puffin remains substantially more affordable than recent general-purpose unified multimodal models such as Bagel (14B) and Qwen-Image (20B).
>
> > ### **W2. Data vs. model contribution.**
>
> Thank you for raising this insightful question. In the revision, we have added experiments where the comparison method (GeoCalib [1]) is re-trained on the same dataset (Puffin-4M) as ours, strictly following its original training recipe. Interestingly, we find that the re-trained GeoCalib on our 4M dataset slightly underperforms the original GeoCalib model trained on its 40K dataset. The evaluation results with median errors on the Puffin-*Und* test set are reported below (the complete results with AUC metrics are listed in Table A6 in the revised paper).
>
> | | **Roll [degrees]** | **Pitch [degrees]** | **FoV [degrees]** |
> | :--- | :--- | :--- | :--- |
> | **GeoCalib (trained on Puffin-4M)** | 1.12 | 2.54| 5.47|
> | **GeoCalib** | 0.92 | 2.18 | 5.04 |
> | **Puffin** | 0.41 | 0.74 | 1.21 |
>
> By carefully analyzing these results and revisiting the technical details of GeoCalib, we offer two explanations for this phenomenon:
>
> - **Model capacity *vs.* data scale.** GeoCalib is a relatively lightweight CNN-like architecture with ~29M parameters. When trained on a significantly larger 4M-scale dataset with broad scene and distribution coverage, it tends to *underfit*: its limited capacity cannot fully model the entire distribution, so it only fits some sub-distributions well while inevitably neglecting others.
>
> - **Consistent observations in the original paper.** The GeoCalib authors report a similar trend in their ablation: when training the network on a 5x larger dataset, the camera understanding performance *slightly degrades* on most benchmarks, and no clear improvement is observed; they also show that more advanced architectures can further boost performance. These observations are fully consistent with our findings.
>
> In contrast, Puffin is built on a high-capacity LLM backbone, which, like other large multimodal models (*e.g.*, LLaVA, Qwen-VL, InternVL), requires sufficiently large and diverse training data to avoid overfitting and to learn an accurate joint distribution over images, camera geometry, and language. In this sense, Puffin-4M is not merely a “bonus”, but rather a *necessary data regime* for such a unified multimodal model that supports both understanding and generation.
>
> Furthermore, we also conducted experiments by training our model on the Puffin-4M dataset of different scales. As shown in Figure A6(b) in the revised paper, we observe clear and consistent improvements as the data scale increases. Overall, these results suggest that the dataset and the model contribute jointly and should not be viewed in isolation. Following your valuable suggestion, we have added the above results and the corresponding analysis to the revised paper.
>
> > ### **W3. Multi-round conversation capability.**
>
> We agree that exploring the multi-round conversational capability of Puffin is important. As suggested, we have added experiments on multi-turn interleaved conversations (generation → understanding and understanding → generation) in the revised paper. As shown in Figure A7, Puffin can carry out coherent interleaved dialogues conditioned on previous turns. Specifically, it produces consistent cross-view generation results based on its previous reasoning, and accurate camera understanding based on its own generated images. This demonstrates that Puffin not only supports both capabilities within a single framework, but can also use them in an interactive way over multiple conversational rounds, without any task-specific switching or separate models.

---

> ### Author Response · Authors · 2025-11-21
> **Response to Reviewer pAbj (Part 2/2)**
>
> > ### **Q1. Mutual benefit between understanding and generation.**
>
> This is a very interesting question, and we would like to address it from the following two perspectives.
>
> - As shown in Figure A5 in the Appendix, we have shown the mutual effect between camera-centric understanding and generation tasks quantitatively. In particular, isolating the camera understanding component underperforms the unified framework, as the camera-controllable generation branch provides auxiliary diffusion loss that sharpens *fine-grained geometric perception*. Such low-/mid-level knowledge cannot be fully captured by the language model alone. Although the gain for camera-controllable generation is smaller than for camera understanding in Figure A5(b)(d), we still see clear improvements in challenging cases such as FoV simulation, which requires precise and holistic spatial reasoning.
>
> - For other general-purpose unified models, the understanding tasks mainly target high-level concepts such as recognition and semantic comprehension. As a result, the domain gaps across multimodal tasks are more pronounced in these models, likely requiring more delicate architectural designs to harmonize representations across different modalities. We leave a deeper investigation of this interesting direction to future work.

---

### Author Response · Authors · 2025-11-21
**Global Response**

Dear ACs and Reviewers,

We would like to thank all reviewers for providing constructive feedback that helped us improve the paper. We are encouraged that **all reviewers agree this work offers good contribution and soundness to research fields** and acknowledge our paper:

“unifying the camera understanding and controllable generation in one framework by thinking with camera is **novel, meaningful, impactful, and interesting.** It positions as a step toward **foundation-level models that natively incorporate 3D geometry** in multimodal tasks.” (Reviewer pAbj, Reviewer j8cQ, Reviewer 2Lvj, Reviewer D4GA)

“extensive experiments on benchmarks demonstrate **Puffin’s strong performance across both understanding and generation tasks.**” (Reviewer pAbj, Reviewer j8cQ, Reviewer 2Lvj, Reviewer D4GA)

“the carefully constructed **Puffin-4M dataset will be a great contribution and be very helpful to the community.**” (Reviewer pAbj, Reviewer j8cQ, Reviewer 2Lvj)

“the paper is **technically dense, well-motivated, well-structured, and the presentation is excellent.**” (Reviewer pAbj, Reviewer j8cQ, Reviewer 2Lvj, Reviewer D4GA)

We have been working diligently on improving the paper on several fronts, addressing your critique. We hope the answers align well with the reviewers’ concerns. Please see our reviewer-specific feedback for more information as follows.

---

### Author Response · Authors · 2025-12-03
**Summary of Reviewers’ Responses and Discussion**

Dear Reviewers, Area Chairs, Senior Area Chairs, and Program Chairs,

We would like to sincerely thank you for your thoughtful comments, and we especially appreciate your extra efforts in handling our paper given the recent OpenReview incident.

We would like to briefly clarify the situation regarding the reviewer feedback prior to the incident. Before the OpenReview issue occurred, Reviewers j8cQ, 2Lvj, and D4GA had already interacted with our rebuttal and indicated that their concerns were satisfactorily addressed. Reviewer D4GA additionally raised a minor writing issue regarding the reference citation format, and we have carefully corrected all such instances in the revised manuscript.

Unfortunately, the only reviewer who did not have the opportunity to join the rebuttal was Reviewer pAbj, for whom we had provided a detailed, point-by-point response on November 21, 2025. In that response, we added experiments where we retrained comparison methods on the same dataset and analyzed how our model scales with data size. We believe these new results address the concerns raised by Reviewer pAbj. Since similar questions about performance scaling were raised by other reviewers (Reviewer 2Lvj) and later confirmed to be resolved after our rebuttal, we are cautiously confident that Reviewer pAbj’s related concerns have also been properly addressed.

We are grateful to all reviewers for their insightful and constructive feedback, and we sincerely appreciate the hard work of the ACs, SACs, and PCs in serving the community this year.

Best regards,

The Authors

---

### Meta-Review · Area_Chair_gwqS · 2025-12-12

**Summary:**

The paper focuses on developing a unified multimodal model for camera-centric understanding and generation. The initial reviews are overall positive. Reviewers raised concerns regarding the lack of analysis of computational cost, the specific contributions of the data and proposed model, multi-round conversational capability, mutual benefits between understanding and generation, the need for additional ablation studies on the camera-parameter tokenizer, the effects of data (size, quality, and captions), the impact of reinforcement learning methods, model robustness, and minor typos. The authors provided detailed rebuttal responses that addressed most of these concerns. After the rebuttal, three reviewers (j8cQ, 2Lvj, and D4GA) confirmed their original ratings. The AC agreed with the reviewers’ opinions and recommended acceptance of the paper.

**Reviewer Concerns:**

Most of concerns have been addressed and reviewer j8cQ, 2Lvj, and D4GA have confirmed their original ratings. And the authors responses have also address the concerns raised by Reviewer pAbj by providing analysis of computational cost, additional results of training GeoCalib on the proposed dataset, explanation of multi-round conversational capability, mutual benefits between understanding and generation.

**Reviewer Scores:**

All initial reviewer scores are positive (two accepts and two borderline accepts) and reviewer j8cQ, 2Lvj, and D4GA have confirmed their original ratings. I think Reviewer reviewer pAbj would have maintained or raised the score after discussion.

---

### Decision · Program_Chairs · 2026-01-26

Accept (Poster)